# Oversmoothing, "Oversquashing", Heterophily, Long-Range, and more: Demystifying Common Beliefs in Graph Machine Learning

**Adrian Arnaiz-Rodriguez**[*]
ELLIS Alicante
adrian@ellisalicante.org

**Federico Errica**[*]
NEC Laboratories Europe
federico.errica@neclab.eu

## Abstract

After a renaissance phase in which researchers revisited the message-passing paradigm through the lens of deep learning, the graph machine learning community shifted its attention towards a deeper and practical understanding of message-passing's benefits and limitations. In this paper, we notice how the fast pace of progress around the topics of oversmoothing and oversquashing, the homophily-heterophily dichotomy, and long-range tasks, came with the consolidation of commonly accepted beliefs and assumptions – under the form of universal statements – that are not always true nor easy to distinguish from each other. We argue that this has led to ambiguities around the investigated problems, preventing researchers from focusing on and addressing precise research questions while causing a good amount of misunderstandings. Our contribution is to make such common beliefs explicit and encourage critical thinking around these topics, refuting universal statements via simple yet formally sufficient counterexamples. The end goal is to clarify conceptual differences, helping researchers address more clearly defined and targeted problems.

## 1 Introduction

The last decade has seen an increasing scholarly interest in machine learning for graph-structured data (Sperduti & Starita, 1997; Micheli & Sestito, 2005; Gori et al., 2005; Bacciu et al., 2020). After an initial focus on the design of various message-passing architectures (Gilmer et al., 2017), inheriting from the recurrent (Scarselli et al., 2009) and convolutional (Micheli, 2009) Deep Graph Networks (DGN), together with the analysis of their expressive power (Xu et al., 2019; Morris et al., 2023), researchers later turned their attention to the intrinsic limitations of the message-passing strategy and the relation between the graph, the task, and the attainable performance. We refer, in particular, to the fact that node embeddings may become increasingly similar to each other as more message-passing layers are used (Rusch et al., 2023), the loss of information that results from aggregating too many messages onto a single node embedding (Alon & Yahav, 2021), the presence of topological bottlenecks (Topping et al., 2022), the existence of neighbors of different classes (Wang et al., 2024), and the propagation of information between far ends of a graph (Dwivedi et al., 2022). Addressing these limitations makes a difference when applying message-passing models, including foundational ones (Beaini et al., 2024), to large and topologically varying graphs at different scales, from proteins with hundreds of thousands of atoms (Zhang et al., 2023) to dynamically evolving social networks (Longa et al., 2023) with billions of users, where such limitations manifest together.

A pace of research so rapid can sometimes lead, however, to the premature consolidation of ideas and beliefs that have not been thoroughly verified. There are several reasons for this to happen: the (perhaps too) intense pressure to publish, follow the latest scientific trends, and demonstrate state-of-the-art performance. As a result, we may end up putting the spotlight on positive findings but overlooking contradictory evidence, eventually accepting hypotheses as canon.

---

[*]Equal Contribution

Table 1: List of common beliefs. For a non-exhaustive list of papers that make those claims, please refer to Table 2 in the Appendix.

| | Beliefs | | Beliefs |
|---|---|---|---|
| **OSM** | 1. OSM is the cause of performance degradation.
2. OSM is a property of all DGNs. | **OSQ** | 6. OSQ synonym of a topological bottleneck.
7. OSQ synonym of computational bottleneck. |
| **Hom-Het** | 3. Homophily is good, heterophily is bad.
4. Long-range propagation is evaluated on heterophilic graphs.
5. Different classes imply different features. | | 8. OSQ problematic for long-range tasks as well.
9. Topological bottlenecks associated with long-range problems. |

In this paper, we argue and provide evidence that this is potentially the case for the afore-mentioned issues of *oversmoothing* (OSM), *oversquashing* (OSQ),[1] *heterophily*, and *long-range dependencies*, driving researchers away by the difficulty of circumventing common beliefs while concurrently introducing novel contributions. Scientific progress is therefore slowed down both in terms of reduced workforce and clarity of the problems to be addressed; failure to acknowledge existing inconsistencies may well lead to a reiterated spreading of questionable claims.

**Contribution:** While reviewing the literature, we identified and grouped together **nine common beliefs** that cause great confusion and ambiguities in the field, especially when taken separately. We then demystify such beliefs by providing *simple* counterexamples that should be easy to recall. By encouraging critical thinking around these issues and separating the different research questions, we hope to foster further advancements in the graph machine learning field.

*Disclaimer:* We remark that the goal of this paper is not explicitly pinpointing criticalities in previous works; on the contrary, these works were fundamental to forming our current understanding and ultimately producing this manuscript. We also acknowledge that the list of referenced works cannot be exhaustive, due to the field's size, and it mainly serves to support our arguments.

Table 1 summarizes our findings about common beliefs in the literature, together with the list of papers where we could find mentions of them.[2] We logically divide common beliefs about OSM, OSQ, and the homophily-heterophily dichotomy. In the following sections, we discuss each belief, provide counterexamples, and summarize our arguments with take-home messages.

For readers that are less familiar with the topic and its definitions, Appendix B.1 provides a brief introduction to message-passing models. In addition, Appendix E provides a discussion with alternative perspectives on the issues discussed in this paper. The code we use to run experiments is available at `https://github.com/AdrianArnaiz/demystifyingGraphML`.

## 2 IS OVERSMOOTHING A REAL ISSUE?

Oversmoothing broadly refers to the phenomenon where, as we stack more message-passing layers in a DGN, the node embeddings become increasingly similar to each other, eventually collapsing into a low-dimensional (Huang et al., 2020; Oono & Suzuki, 2020)–or even single-vector (Cai & Wang, 2020)–subspace. This creates an almost constant representation, independent of the original node-feature distribution, and can **potentially** result in loss of discriminative power (Li et al., 2018; Cai & Wang, 2020; Oono & Suzuki, 2020).

Formally, let $H^\ell \in \mathbb{R}^{n \times d}$ be the matrix of node embeddings after $\ell$ message-passing layers in a DGN, where $n$ is the number of nodes and $d$ is the hidden dimension. Consider a similarity (or separation) function $\pi \colon \mathcal{H} \to \mathbb{R}$, where $\mathcal{H} = \mathbb{R}^{n \times d}$ is the space of all possible embedding matrices. We say that a DGN experiences OSM if

$$\lim_{\ell \to \infty} \pi(H^\ell) = c. \qquad (1)$$

---

[1] We discuss and address the troubling abuse of the term "*oversquashing*" in later sections.

[2] Sometimes we found overly general claims in the first sections, later refined, which nonetheless contribute to the spreading of common beliefs.

where $c$ is some constant indicating a collapse of embeddings. Deviation metrics (e.g., Dirichlet Energy (Cai & Wang, 2020), MAD (Chen et al., 2020a)) and subspace-collapse criteria (Oono & Suzuki, 2020; Huang et al., 2024; Roth & Liebig, 2024) all measure the same intuition: as depth increases, node embeddings shrink toward a nearly constant subspace or degree-proportional vectors.

## 2.1 Belief: OSM is a Property of All DGNs.

A widespread claim in the literature is that *OSM happens regardless of the specific architecture or the underlying graph*. Early theoretical works support this view by analyzing message-passing propagation as a diffusion process: iterated normalized-Laplacian updates converge to a degree-weighted stationary distribution (Cai & Wang, 2020; Giraldo et al., 2023), while heat-kernel diffusion converges to a constant vector (Oono & Suzuki, 2020; Arnaiz-Rodriguez & Velingker, 2024). The resulting bounds quantify the rate of OSM in terms of the singular values of the feature transform $W$ and the eigenvalues of the graph structure $G$.

These conclusions, however, rely on restrictive assumptions. Later work has relaxed them by introducing learnable feature transforms, non-linear activations, and more elaborate architectures (Oono & Suzuki, 2020; Cai & Wang, 2020; Wu et al., 2023a), yet no existing proof shows inevitable collapse under realistic training regimes. In practice, remedies such as residual/skip connections, normalization layers, or gating mechanisms are explicitly architectural changes designed to maintain local distinctions, calling into question the universality of this OSM claim.

In addition, many studies probe OSM with *untrained* (weights frozen at initialization) linear GCN stacks, an experimental choice that hides the effect of learning and may lead to the wrong conclusions, as noted by Zhang et al. (2025). Indeed, Cong et al. (2021, Figure 2) report OSM only for frozen-weight networks; once parameters start adapting, the models preserve informative variance. Together, these observations suggest that OSM is not an inevitable consequence of message-passing but rather a contingent outcome that depends on training dynamics and architectural design choices.

**Empirical Example** We show a simple training scenario where we see how OSM is *not* a property of all DGNs and how different elements make it difficult to draw clear conclusions. We used several DGNs under two propagation variants: the vanilla $AXW$ update and the rescaled $AX(2W)$, following the same experimental procedure as Roth & Liebig (2024, Figure 1). We measure OSM with 2 different metrics: Dirichlet Energy (DE) and its norm-normalized version, the Rayleigh Coefficient (RQ), which was also previously used in some works (Cai & Wang, 2020; Di Giovanni et al., 2023b; Roth & Liebig, 2024; Maskey et al., 2023), defined as:

$$\mathrm{DE}(\mathbf{H}^\ell) = \mathrm{Tr}\big((\mathbf{H}^\ell)^T \mathbf{L} \mathbf{H}^\ell\big) = \frac{1}{2} \sum_{u,v \in \mathcal{E}} \|\mathbf{h}_u - \mathbf{h}_v\|_2^2, \qquad \mathrm{RQ} = \frac{\mathrm{Tr}\big((\mathbf{H}^\ell)^T \mathbf{L} \mathbf{H}^\ell\big)}{\|\mathbf{H}^\ell\|_F^2} \qquad (2)$$

where $\mathbf{L}$ is the graph Laplacian and $\mathbf{H}^\ell$ is the node embedding matrix at layer $\ell$. DE measures the raw smoothness of the embeddings, while RQ normalizes this smoothness by the overall scale of the embeddings, providing a relative measure of how much the embeddings are collapsing compared to their norm. Appendix B.2 provides more details on these metrics and their relation to OSM.

Figure 1 shows three key facts: (i) some architectures never collapse, (ii) a minor rescaling can reverse the trend, and (iii) DE and RQ often disagree. Hence, OSM is neither universal nor straightforward to diagnose.

First, OSM, as measured by DE, is not universal: GIN's DE explodes instead of collapsing in the vanilla setting (a), which shows that changing the aggregation-function may lead to an opposite behavior of the OSM effect.

Second, a minor scaling in the feature transformation ($2W$ instead of $W$) flips the behavior of several models: curves that decayed in (a) now grow or stabilize, and vice-versa; this behavior is theoretically analyzed in Di Giovanni et al. (2023b). Similar small tweaks (normalization layers, self-loops, alternative aggregators) can likewise create or remove DE collapse, which has been leveraged by prior work to propose OSM mitigation approaches, such as the ones based on feature normalization (Wu et al., 2019; Zhao & Akoglu, 2020; Zhou et al., 2020).

Finally, whether OSM is observed depends on the measure of choice. DE reflects raw smoothness, whereas the RQ normalizes by the feature norm. As a result, the same model can exhibit mutually

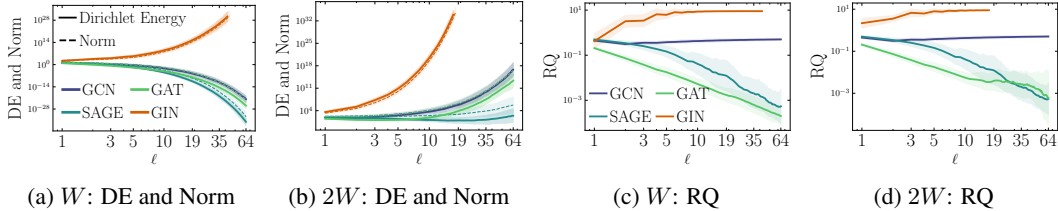

(a) $W$: DE and Norm     (b) $2W$: DE and Norm     (c) $W$: RQ     (d) $2W$: RQ

Figure 1: **(a-b)**: We depict the evolution, with increasing number of layers, of the DE, using $W$ and $2W$ feature transformations for different architectures. **(c-d)**: Evolution of the RQ for $W$ and $2W$ as before. Experiments run on the Cora dataset for 50 random seeds. A larger version for better visualization is available in Fig. 8.

contradictory trends for different metrics (Zhang et al., 2025). First, GCN's DE collapses under the vanilla aggregation (a), explodes with a simple rescaling (b), yet RQ remains essentially flat in both normalised plots (c–d), indicating that GCN embeddings are being rescaled, not necessarily oversmoothed. On the contrary, GAT and SAGE, which had similar behavior as GCN in (a) and (b), decay using RQ (c-d) at a (roughly) linear rate, highlighting how different architectures respond differently to the use of RQ instead of DE. Furthermore, subfigure (d) shows that $AX(2W)$ can stabilize RQ for certain methods, suggesting that normalizations or small parameter adjustments do not affect all models uniformly. Therefore, conclusions about OSM depend heavily on metric and model, an observation that we rarely found in the literature.

In conclusion, OSM is neither inevitable nor uniquely defined: its observation hinges on the architecture, on seemingly innocuous hyper-parameters, and on which stability metric (DE vs. RQ) one adopts (Zhang et al., 2022). Therefore, a natural question arises: Does OSM actually limit the models' predictive accuracy? We investigate this question in the next section.

## 2.2 Belief: OSM is the Cause of Performance Degradation.

Part of the literature focuses on the narrative that OSM is the cause of lower test accuracy in DGNs. The hypothesis is that if embeddings collapse to a non-meaningful space, then the separability of the nodes will become challenging, and accuracy decreases.

However, this hypothesis ignores some critical aspects, such as *i)* the separability of node embeddings with respect to the nodes' labels, and *ii)* how such separability evolves in the intermediate OSM phase (*if* it happens, as we discussed before).

Regarding the first statement, although it is true that if there is total collapse to the same value then the embeddings will not be informative at all, the main problem remains the node embeddings' separability. As shown in the previous section, some changes in the architectures can avoid OSM, but they might have no impact on the overall accuracy. For instance, multiplying by two the weight matrix leads to a general increase in DE for all architectures; however, the accuracy will remain the same since the embeddings have been simply scaled up, and the embeddings' separability is not affected negatively. In addition, avoiding an embedding collapse does not necessarily lead to an improvement in generalization accuracy. For instance, comparing a GCN with and without bias, both versions show a decrease in performance as the number of layers increases, whereas only GCN shows a collapse in DE (Rusch et al., 2023, Figure 3).

On the other hand, and related to the second statement, embedding collapse will not always lead to a decrease in accuracy. OSM happens faster in some subspaces than in others, and this effect will be beneficial if labels are correlated with those subspaces (Keriven, 2022). For instance, if we classify points into two classes, and all nodes of distinct classes collapse into different points, the OSM metric will detect such a collapse. However, the separability of the node embeddings will remain possible, illustrating also the limits of widely used OSM metrics with respect to label information.

This intuitive behavior has been identified in the literature as a form of "beneficial" smoothing phase (Keriven, 2022; Roth & Liebig, 2024; Wu et al., 2023b). In this phase, the nodes of each class first collapse into a class-dependent point before the *potential* second stage, at which point all nodes converge to the same representation. Finally, although overall pairwise distances might shrink in

deep layers, *within-class* distances might contract more than *between-class* ones, so class separability improves despite the global collapse, as discussed by Cong et al. (2021).

In conclusion, low accuracy in DGNs cannot be attributed to OSM alone. The separability of node embeddings plays a major role, where other training problems such as vanishing gradients or over-fitting also arise when using a big number of layers (Zhao & Akoglu, 2020; Cong et al., 2021; Yang et al., 2020; Arroyo et al., 2025; Park et al., 2025).

> **Call to Action**
>
> i) Do not assume OSM applies to all DGNs
>
> ii) Avoid attributing performance drops to OSM by default. The performance is related to node embeddings' separability, which can also be affected by many other elements, such as vanishing gradients or over-fitting.
>
> iii) Shift research towards achieving separability of node embeddings and, optionally, how OSM relates to that.

## 3 HOMOPHILY-HETEROPHILY AND THE ROLE OF THE TASK

In the context of node classification, the term *homophily* (resp. *heterophily*) generally refers to some form of similarity (resp. dissimilarity) between a node and its neighbors (McPherson et al., 2001). This (dis)similarity can be measured with respect to class labels, node features, or both; the vast majority of works in the literature opt for the first, *but this choice is often implicit and taken for granted*, making some statements hard to interpret when one is aware of the other ways to measure it.

### 3.1 BELIEF: HOMOPHILY IS GOOD, HETEROPHILY IS BAD

A recurrent narrative in the literature is that the message-passing mechanism of DGNs is particularly suited for homophilic graphs, whereas it is unfit for heterophilic graphs. The apparent motivation is that, in homophilic graphs, all you need to do to solve a node classification task is to look at similar neighbors, and a local message-passing strategy implements just the right inductive bias. This is in contrast to a class-heterophilic graph, where there exist neighboring nodes of a different class that might make it harder for message-passing to isolate the "relevant information", intended as neighbors of the same class. Such a belief is supported by empirical evidence on a rather restricted set of benchmarks (Sen et al., 2008; Namata et al., 2012; Pei et al., 2020) with varying levels of homophily/heterophily.

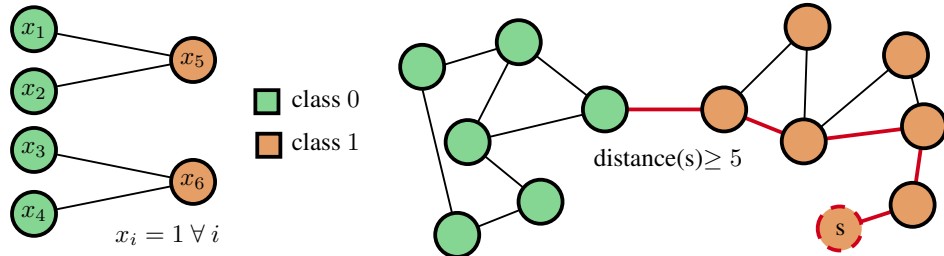

Figure 2: *Left:* a fully heterophilic graph inspired by Ma et al. (2022) where a 1-layer, sum-based DGN can perfectly classify the nodes due to a difference in the node degree. *Right:* a highly homophilic graph where the task is to predict if a node is at a distance greater than five from a specific node. Here, the performances of a DGN will be poor unless information from nodes of another community – from the perspective of a class-0 node – is captured.

Researchers already tried to challenge these considerations in the past (Ma et al., 2022; Errica, 2023; Luan et al., 2023; Platonov et al., 2023b; Wang et al., 2024). Consider Figure 2 (left), where a bipartite graph with identical node features is fully class-heterophilic. If we apply a single sum-based graph convolution, the nodes can be perfectly classified as the resulting embeddings depend on the incoming degree. Therefore, **there exist heterophilic graphs where a DGN can achieve perfect classification**.

On the contrary, Figure 2 (right) depicts a highly class-homophilic graph, where nodes belong to one of two classes if they are at a distance greater or lower than five from a given node $s$. In this case, the information on the nodes does not even matter; if we were to follow the above belief, we would be encouraged to use a few layers of message-passing, and as a result **we could never solve the task perfectly**. These considerations have motivated the definition of new metrics that put homophily and heterophily in relation with performance under certain assumptions, for instance in Ma et al. (2022).

## 3.2 Belief: Different Classes Imply Different Features

In the previous section, we deliberately ignored the interplay between a node's features and its class label, which is induced by the task at hand. The reason is that we wanted to clarify the distinction with another, more subtle, and problematic belief: nodes belonging to different classes should have different (*i.e.*, separable) feature distributions. Under this assumption, class homophily implies feature homophily.

Such an assumption is often key in arguments supporting the belief of Section 3.1. Indeed, if nodes of different classes have different feature distributions, then applying a local message-passing iteration to a highly class-homophilic graph should "preserve the distance" between node embeddings of different classes. On the contrary, in a heterophilic setting, a graph convolution would mix information coming from different feature distributions, which may be detrimental to performance.

The logic is not incorrect per-se, but our key counterargument is the following: if different classes imply different feature distributions, why would one need to apply a DGN rather than a simple MLP? In other words, **either there is a very strong assumption** that the task does not depend on the topological information, or the feature-class distributions induced by the task allow us to somehow take a shortcut in terms of learned function, neglecting the role that the topology might have. Note that this discussion is irrespective of the denoising/regularizing effects of DGNs in semi-supervised scenarios compared to MLPs (Hoang et al., 2021; Errica, 2023).

It appears therefore necessary to consider less trivial and more fine-grained scenarios, where the feature distributions of different classes partially or totally overlap, the topology has a key role in the task definition, and topological properties induce a positive/negative effect on the performance of message-passing models as done, for instance, in recent works (Castellana & Errica, 2023; Zheng et al., 2024).

## 3.3 Belief: Long-range Propagation is Evaluated on Heterophilic Graphs

The common beliefs of Sections 3.1 and 3.2 have been used to support yet another argument, namely that we should evaluate the ability of DGNs to propagate long-range information on heterophilic graphs. The rationale seems to be that, in order for DGNs to perform well, nodes of a given class should focus on information of similar (w.r.t. class and/or features) nodes; therefore, in a heterophilic graph, it may be necessary to capture information far away (i.e., long-range) from the immediate neighborhood.

Once more, what is really important is to **distinguish the task**, e.g., one that depends on long-range propagation, **from the class labels the task induces on the nodes**. As a matter of fact, the "long-range" task of Figure 2 (right) induces a highly homophilic graph, while the heterophilic graph of Figure 2 (left) is not associated with a long-range propagation task. Therefore, we cannot draw a generic relation between long-range tasks and heterophily without making further assumptions.

> **Call to Action**
>
> i) Do not rely on generic claims about the performance of DGNs under homophily and heterophily, nor about their relation with long-range problems.
>
> ii) Treat homophily/heterophily properties as task-dependent, not the converse.
>
> iii) Move past the coarse homophily-heterophily dichotomy and **focus more on the task** and the interplay between features, structure, and class labels.

## 4   THE MANY FACETS OF "OVERSQUASHING" AND THEIR NEGATIVE IMPLICATIONS

The term oversquashing originated from Alon & Yahav (2021) and referred to an "*exponentially growing information into a fixed-size vector*" by repeated application of message-passing. In other words, oversquashing was associated with the **computational tree** (Figure 3) induced by message-passing layers on each node of the graph. Later, oversquashing was connected by Topping et al. (2022) to the existence of **topological bottlenecks**: "*edges with high negative curvature are those causing the graph bottleneck and thus leading to the over-squashing phenomenon*". Since then, researchers have adopted one or even both definitions of oversquashing at the same time, contributing to an apparent understanding that these definitions subsume the same problem.

In this section, we argue that **this is not the case** and that, as a community, **we should clearly separate the term "oversquashing"** into *(at least)* two separate terms:

**Computational Bottlenecks**   and   **Topological Bottlenecks**

Computational bottlenecks, defined in Def. 4.1, are inevitably related to the *message-passing architecture*, for which the graph is the computational medium, whereas topological bottlenecks refer to the *graph connectivity*, usually measured with spectral or curvature metrics defined in Appendix B.3.2. Therefore, the topological bottlenecks are intrinsic to the graph, while the computational bottleneck is intrinsic to the architecture or procedure. While these bottlenecks are clearly intertwined, in the following we show that they do not always coexist, hence it makes sense to treat them as **fundamentally distinct concepts**.

***Computational* and *Topological* Bottleneck definitions.**   Although both bottlenecks have been previously used to measure "*oversquashing*", their definitions and measurement approaches also differ across the literature. There is no universally accepted formal definition of a *topological* bottleneck; instead, it is typically characterized using proxy metrics, most often involving the spectral gap of the curvature. Some of the most significant metrics are summarized in B.3.2.

On the other hand, the *computational* bottleneck has been usually measured using the *receptive field* (Chen et al., 2018; Alon & Yahav, 2021), $\mathcal{N}_v^k$, as the set of $K$-hop neighbors, which grows exponentially with the number of layers.

However, when evaluating the actual *computational* graph produced by message-passing, shown in Figure 3 (right), duplicate nodes become significant: a node appearing in multiple branches of the computation tree contributes repeatedly, as each unique walk generates a distinct message. Consequently, we employ multiset notation to formally define the *computational tree* for a node $v$:

$$\mathcal{M}_v^1 := \mathcal{N}_v, \quad \mathcal{M}_v^K := \mathcal{M}_v^{K-1} \uplus \left\{ \biguplus_{u \in \mathcal{M}_v^{K-1}} \mathcal{N}_u \right\}. \tag{3}$$

Therefore, we can define the notion of an "exponentially-growing receptive field" (Alon & Yahav, 2021) as follows:

**Definition 4.1** (Computational Bottleneck). For a given node $v$ and number of message passing layers $K$, the computational bottleneck of node $v$ is defined as $|\mathcal{M}_v^K|$.

We refer the reader to B.4 for further insights, related work, and connections between this definition and matrix-power interpretations of message passing.

Figure 3 visualises the difference between the set size $|\mathcal{N}_v^K|$ and the multiset size $|\mathcal{M}_v^K|$ on a toy graph and on a simple graph.

### 4.1   BELIEF: OVERSQUASHING AS SYNONYM OF TOPOLOGICAL BOTTLENECK

The prolific line of work that associates "oversquashing" with topological bottlenecks seems to have gained popularity with Topping et al. (2022). In that paper, edges with negative curvature are first associated with (topological) bottlenecks, then a theorem puts in relation message-passing

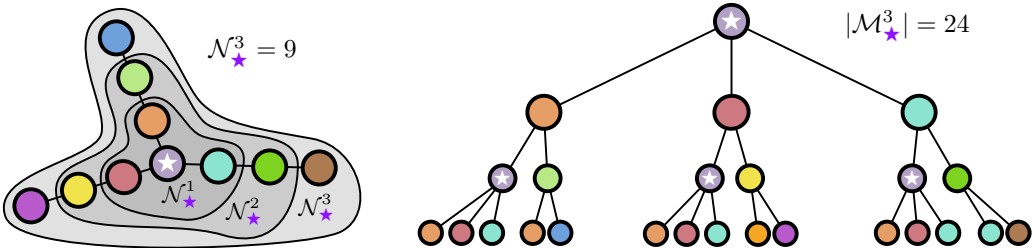

Figure 3: We intuitively visualize what happens when we repeatedly aggregate the neighborhood of the star node using the message-passing paradigm, where we define the computational bottleneck as the size of the computational tree (24 computational nodes) for $k = 3$ *vs* the receptive field for $k = 3$ that includes 9 three-hop neighbors).

on a graph containing a bottleneck with the Jacobian sensitivity of node representations, typically defined by $I(u, v) = \|\partial \mathbf{h}_u^K / \partial \mathbf{h}_v^0\|$ and denoting how much the final representation of a node $u$ after $K$ layers is influenced by the initial representation of a node $v$ (Xu et al., 2018). Put simply, a topological bottleneck may imply low sensitivity. **However, the converse is not necessarily true:** we can have low sensitivity on a graph where there are no bottlenecks. Figure 4 (left) shows a grid graph where there are no topological bottlenecks. The repeated application of message passing will, however, quickly generate a computational bottleneck. Therefore, saying that there are no topological bottlenecks does not imply that there are no computational bottlenecks.

To improve on topological bottlenecks, a widely investigated approach is graph rewiring (Attali et al., 2024b), which was also the subject of scrutiny recently (Tortorella & Micheli, 2022; Tori et al., 2025). Rewiring is based on the intuition that improving topological bottlenecks metrics should improve the performance of DGNs (Arnaiz-Rodriguez et al., 2022; Banerjee et al., 2022; Karhadkar et al., 2023; Deac et al., 2022), by reducing the distance between nodes that should communicate. At the same time, it should become clear now that, under the DGN paradigm, *rewiring might worsen the computational bottleneck* – as long as the same number of message-passing layers is used – while improving the topological one. This perspective was also put forward by Errica et al. (2025), with a theoretical analysis on how message filtering,[3] as shown in Figure 4 (middle), reduces both the computational bottleneck and sensitivity yet improves performances while leaving the graph structure unaltered.

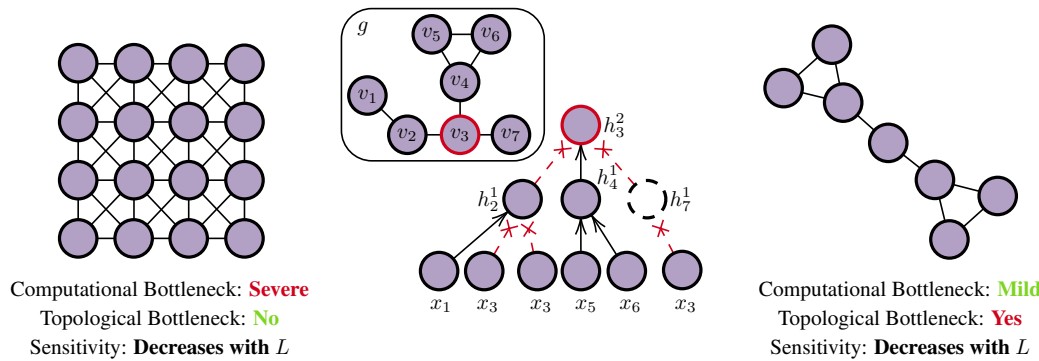

Figure 4: *Left:* in a grid graph, the computational bottleneck grows very quickly, but there is no topological bottleneck. *Middle:* A visualization of the computational graph rooted at node $v_3 \in \mathcal{V}_g$ for two message passing layers, highlighting how pruning messages reduces the computational bottleneck. *Right*: In this graph, there is a topological bottleneck and a mild computational bottleneck. As with the grid graph (Appendix C), the sensitivity decreases with the number of message-passing layers.

---

[3]The concept of *message filtering* refers to message passing architectures with the ability to learn how many messages to exchange between nodes and which messages to filter out.

Another, slightly more technical way to see why low Jacobian sensitivity does not imply the presence of any topological bottlenecks is to follow the chain of upper bounds that link the metrics used to measure the computational and topological bottlenecks (Black et al., 2023; Di Giovanni et al., 2023a; Karhadkar et al., 2023; Arnaiz-Rodriguez et al., 2022).

Several recent works (Black et al., 2023; Di Giovanni et al., 2023a) have shown that sensitivity is upper bounded by a term that includes the effective resistance, a purely topological distance metric that quantifies the expected commute time of a random walk between nodes $u$ and $v$ (Klein & Randić, 1993). In particular, the bound subtracts the lower bound on the maximum effective resistance, which depends on the inverse of the spectral gap –a proxy for topological bottlenecks– (Chandra et al., 1989; Lovász, 1993). This connection provides a useful intuition: as the topological bottleneck gets worse, the lower bound on the maximum effective resistance increases, which in turn reduces the upper bounds on the sensitivity of Black et al. (2023).

**However, the converse does not hold.** There are graphs where the maximum effective resistance between distant nodes is large despite the absence of topological bottlenecks. For instance, consider the example of a grid graph in Figure 4 (left) where there is no topological bottleneck, yet the effective resistance between diagonally opposite corners grows linearly with the grid size. As a result, sensitivity between those nodes decays with depth, even though the graph has no identifiable topological bottlenecks. This illustrates that computational bottlenecks can arise independently of topological ones, **and that low sensitivity does not necessarily imply the presence of either of them**.

## 4.2 BELIEF: OVERSQUASHING AS SYNONYM OF COMPUTATIONAL BOTTLENECK

We briefly complement the previous section with a discussion on "oversquashing" as a computational bottleneck, which was introduced by Alon & Yahav (2021) and has been the (often implicit) study subject of works that prune, to some extent, the computational tree induced on every node by the iterative message-passing process (Rong et al., 2020; Errica et al., 2025). Also in this case, there exist cases where reducing the computational bottleneck may be harmful: Figure 4 (right) shows a graph where there is a topological bottleneck but no severe computational bottleneck (for a limited number of layers). In this case, excessive pruning of the computational tree might cause distant nodes to interrupt all communications. Therefore, computational and topological bottlenecks are problems that should be tackled separately.

## 4.3 BELIEF: OVERSQUASHING IS PROBLEMATIC FOR LONG-RANGE TASKS AS WELL

Since its definition by Alon & Yahav (2021), oversquashing has often been considered a problem in long-range tasks. The reason stems from its relation to the exponentially growing computational tree as the number of message-passing layers increases: whenever a node has to receive information from another node at distance $d$, classical (synchronous) message-passing architectures need to apply at least $d$ layers to capture that information. As a result, the relevant information may get lost due to the exponentially large computational bottleneck. Importantly, topological bottlenecks can only make the problem worse, by forcing the information of a group of messages to be squeezed through an edge – please refer to the next section for a discussion about long-range tasks and topological bottlenecks.

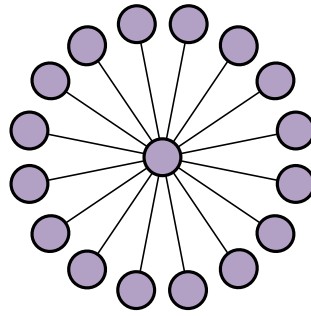

The main message here is that long-range tasks "force" classical message-passing architecture to create a computational bottleneck to propagate the necessary information, **but one can observe computational bottlenecks even in short-range tasks**. An obvious example is the high-degree node of Figure 5, where, after *just one* layer, the center node receives a high number of messages, effectively creating a computational bottleneck in terms of information to be squashed into a fixed-size vector. Therefore, while the task of long-range is related to computational bottlenecks under classical DGNs, computational bottlenecks are not a prerogative of long-range tasks.

Figure 5: Hubs can exhibit computational bottlenecks.

### 4.4 Belief: Topological Bottlenecks Associated with Long-range Problems

The last belief we discuss is that topological bottlenecks are the primary obstacle to solving long-range tasks. This intuition stems from the fact that narrow cuts impede information flow between distant parts of the graph. It is indeed true that a topological bottleneck can worsen communication between distant nodes, especially if the bottleneck lies along a path that connects them. However, this perspective is limited in two important ways.

First, a topological bottleneck is only harmful if it lies on the information paths between nodes that are supposed to communicate. A topological bottleneck may exist without affecting task-relevant dependencies. Second, the graph topology can worsen the long-range communication even in the absence of any identifiable topological bottleneck, by inducing computational bottlenecks. As we discussed in Section 4.1, the grid graph is an illustrative case: despite the lack of topological bottlenecks, to connect the opposite corner nodes we need, at least, as many message-passing layers as the distance between them, thus leading to a huge computational bottleneck that will likely hamper the ability to process long-range dependencies.

In addition, some of the techniques that reduce topological bottlenecks rely on introducing more edges or nodes into the graph, with the aim of reducing the distance between far-away nodes that should communicate. It is important to note that, although these approaches might be beneficial for the task at hand, they also worsen the computational bottleneck by adding more branches to the computational tree.

In conclusion, this highlights a deeper issue: long-range problems are not solely caused by topological bottlenecks, rather they can be understood as a form of information attenuation caused by computational bottlenecks in the message-passing mechanism, which can potentially be exacerbated by topological bottlenecks. Thus, solving a topological bottleneck is neither necessary nor sufficient to solve all long-range problems.

---

**Call to Action**

i) Stop using the ambiguous "Oversquashing", which led to unclear research statements. Talking about *computational* and *topological* **bottlenecks**, instead, better defines the research scope of a paper, since **they are two fundamentally distinct concepts**.

ii) There can be computational bottlenecks but no topological ones, and vice-versa. This also means **we should create ad-hoc benchmarks for each type of bottleneck rather than relying solely on real-world tasks**, where it is not as easy to distinguish the combined effect of the two bottlenecks.

iii) Keep in mind that computational bottlenecks can happen in short-range as well as long-range tasks.

iv) Do not attribute long-range performance issues solely to topological bottlenecks — account for computational bottlenecks as well.

---

## 5 Conclusions

This paper posits that the fast pace of advances in the graph machine learning field has generated several commonly accepted beliefs and hypotheses, rooted in the notions of oversmoothing, "oversquashing", long-range tasks, and heterophily, which are the cause of misunderstandings between researchers. Our contribution was to highlight such beliefs in plain sight, provide an explanation for their emergence, and demystify them when necessary with simple counterexamples. First, we argued that OSM may not be an actual problem and that node embeddings' separability should be preferred when looking for the root causes of performance degradation. Then, we showed how talking about computational and topological bottlenecks resolves most, if not all, inconsistencies generated by the inflated use of the "oversquashing" term. Finally, we highlighted the role of the task in statements involving homophily, heterophily, and long-range dependencies. By providing much-needed clarifications around these aspects, we hope to foster further advancements in the graph machine learning field.

ACKNOWLEDGMENTS

A.A. has been partially supported by a nominal grant received at the ELLIS Unit Alicante Foundation from the Regional Government of Valencia in Spain (Resolución de la Conselleria de Industria, Turismo, Innovación y Comercio , Dirección General de Innovación) and a grant by the Banc Sabadell Foundation.

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

# A    TABLE OF COMMON BELIEFS WITH REFERENCES

Table 2: List of common beliefs together with a non-exhaustive list of papers that make those claims.

| Common Belief and references |
|---|
| **1. OSM is the cause of performance degradation.** |
| (Li et al., 2018; Hamilton, 2020; Cai & Wang, 2020; Chen et al., 2020b;a; Hasanzadeh et al., 2020; Huang et al., 2020; Liu et al., 2020; Rong et al., 2020; Zhao & Akoglu, 2020; Zhou et al., 2020; Wang & Leskovec, 2021; Zhou et al., 2021; Bodnar et al., 2022; Chen et al., 2022; Hwang et al., 2022; Rusch et al., 2022; Tortorella & Micheli, 2022; Cai et al., 2023; Maskey et al., 2023; Nguyen et al., 2023; Karhadkar et al., 2023; Rusch et al., 2023; ud din & Qureshi, 2024; Wu et al., 2023a; Epping et al., 2024; Jamadandi et al., 2024; Roth & Liebig, 2024; Stanovic et al., 2025; Wang et al., 2025) |
| **2. OSM is a property of all DGNs.** |
| (Xu et al., 2018; Gasteiger et al., 2019; Li et al., 2019; Chen et al., 2020b; Hamilton, 2020; Hasanzadeh et al., 2020; Huang et al., 2020; Zhao & Akoglu, 2020; Zhou et al., 2020; Alon & Yahav, 2021; Zhou et al., 2021; Abboud et al., 2022; Arnaiz-Rodriguez et al., 2022; Chen et al., 2022; Hwang et al., 2022; Keriven, 2022; Topping et al., 2022; Rusch et al., 2022; Akansha, 2023; Cai et al., 2023; Di Giovanni et al., 2023a; Errica et al., 2025; Giraldo et al., 2023; Nguyen et al., 2023; ud din & Qureshi, 2024; Rusch et al., 2023; Shao et al., 2023; Wu et al., 2023a;b; Attali et al., 2024a;b; Fesser & Weber, 2024; Jamadandi et al., 2024; Roth & Liebig, 2024; Arroyo et al., 2025; Wang et al., 2025) |
| **3. Homophily is good, heterophily is bad.** |
| (Pei et al., 2020; Zhou et al., 2020; Zhu et al., 2020; Lim et al., 2021; Luan et al., 2021; Wang & Leskovec, 2021; Arnaiz-Rodriguez et al., 2022; Bodnar et al., 2022; Di Giovanni et al., 2023b; Liu et al., 2023b; Platonov et al., 2023a; Bi et al., 2024; Attali et al., 2024a;b; Gong et al., 2024; Roth & Liebig, 2024; Zheng et al., 2024) |
| **4. Long-range propagation is evaluated on heterophilic graphs.** |
| (Arnaiz-Rodriguez et al., 2022; Tortorella & Micheli, 2022; Akansha, 2023; Black et al., 2023; Maskey et al., 2023; Huang et al., 2024; Giraldo et al., 2023; Attali et al., 2024a; Tori et al., 2025) |
| **5. Different classes imply different features.** |
| (Abu-El-Haija et al., 2019; Pei et al., 2020; Lim et al., 2021; Luan et al., 2021; Ma et al., 2022; Sun et al., 2022; Luan et al., 2022; Rusch et al., 2023; Attali et al., 2024a;b; Bi et al., 2024; Huang et al., 2024; Wang et al., 2024; Zheng et al., 2024) |
| **6. OSQ synonym of a topological bottleneck.** |
| (Xu et al., 2018; Arnaiz-Rodriguez et al., 2022; Banerjee et al., 2022; Chen et al., 2022; Deac et al., 2022; Sun et al., 2022; Topping et al., 2022; Tortorella & Micheli, 2022; Akansha, 2023; Balla, 2023; Black et al., 2023; Di Giovanni et al., 2023a; Errica et al., 2025; Gabrielsson et al., 2023; Giraldo et al., 2023; Karhadkar et al., 2023; Liu et al., 2023a; Nguyen et al., 2023; Shao et al., 2023; Shi et al., 2023; ud din & Qureshi, 2024; Yu et al., 2023; Attali et al., 2024a;b; Barbero et al., 2024; Di Giovanni et al., 2024; Fesser & Weber, 2024; Huang et al., 2024; Jamadandi et al., 2024; Southern et al., 2024; Arroyo et al., 2025; Gravina et al., 2025; Stanovic et al., 2025; Tori et al., 2025) |
| **7. OSQ synonym of computational bottleneck.** |
| (Alon & Yahav, 2021; Abboud et al., 2022; Arnaiz-Rodriguez et al., 2022; Banerjee et al., 2022; Chen et al., 2022; Deac et al., 2022; Sun et al., 2022; Topping et al., 2022; Tortorella & Micheli, 2022; Akansha, 2023; Balla, 2023; Black et al., 2023; Dwivedi et al., 2022; Errica et al., 2025; Giraldo et al., 2023; Gutteridge et al., 2023; Nguyen et al., 2023; Karhadkar et al., 2023; Shao et al., 2023; Shi et al., 2023; ud din & Qureshi, 2024; Attali et al., 2024a;b; Barbero et al., 2024; Huang et al., 2024; Southern et al., 2024; Arroyo et al., 2025; Gravina et al., 2025; Stanovic et al., 2025) |
| **8. OSQ problematic for long-range tasks as well.** |
| (Alon & Yahav, 2021; Abboud et al., 2022; Banerjee et al., 2022; Chen et al., 2022; Deac et al., 2022; Topping et al., 2022; Tortorella & Micheli, 2022; Akansha, 2023; Black et al., 2023; Cai et al., 2023; Di Giovanni et al., 2023a; Errica et al., 2025; Gabrielsson et al., 2023; Karhadkar et al., 2023; Nguyen et al., 2023; Liu et al., 2023a; Shi et al., 2023; Yu et al., 2023; Barbero et al., 2024; Di Giovanni et al., 2024; Fesser & Weber, 2024; Huang et al., 2024; Southern et al., 2024; Attali et al., 2024b; Arroyo et al., 2025; Gravina et al., 2025; Stanovic et al., 2025) |
| **9. Topological bottlenecks associated with long-range problems.** |
| (Topping et al., 2022; Tortorella & Micheli, 2022; Akansha, 2023; Black et al., 2023; Karhadkar et al., 2023; Liu et al., 2023a; Shi et al., 2023; Fesser & Weber, 2024) |

Left margin labels: **Oversmoothing** (beliefs 1–2), **Homophily-Heterophily** (beliefs 3–5), **Oversquashing** (beliefs 6–9).

# B  BACKGROUND

## B.1  DEEP GRAPH NETWORKS

We provide a brief excursus into Deep Graph Networks for readers new to the topic.

We can define a graph as a tuple $g = (\mathcal{V}_g, \mathcal{E}_g, \mathcal{X}_g, \mathcal{A}_g)$, with $\mathcal{V}_g$ the set of nodes, $\mathcal{E}_g$ the set of edges (oriented or not oriented) connecting pairs of nodes. $\mathcal{E}_g$ encodes the topological information of the graph and can be represented as an adjacency matrix: a binary square matrix $\mathbf{A}$ where $\mathbf{A}_{uv}$ is 1 if there is an edge between $u$ and $v$, and it is 0 otherwise. Additional node and edge features are represented by $\mathbf{x}_v \in \mathcal{X}_g$ and $\mathbf{a}_{uv} \in \mathcal{A}_g$, respectively. $\mathcal{X}_g$ can be, for instance, $\mathbb{R}^d, d \in \mathbb{N}^+$.

The neighborhood of a node $v$ is the set of nodes that are connected to $v$ by an oriented edge, i.e., $\mathcal{N}_v = \{u \in \mathcal{V}_g | (u, v) \in \mathcal{E}_g\}$. If the graph is undirected, we convert each non-oriented edge into two oriented but opposite ones.

The main mechanism of DGNs is the repeated aggregation of neighbors' information, which gives rise to the spreading of local information across the graph. The process is simple: i) at iteration $\ell$, each node receives "messages" (usually just node representations) from the neighbors and processes them into a single new message; ii) the message is used to update the representation of that node. Both steps involve learnable functions, so DGNs can learn to capture the relevant correlations in the graph.

Most DGNs implement a synchronous message-passing mechanism, meaning each node always receives information from all neighbors at every iteration step. This local and iterative processing is at the core of DGNs' efficiency since computation can be easily parallelized across nodes. In addition, being local means being independent of the graph's size. When one learns the same function for all message passing iterations, we talk about *recurrent* architectures, as the GNN of Gori et al. (2005); Scarselli et al. (2009); on the contrary, when one learns a separate parametrization for a finite number of iterations (also known as layers), we talk about *convolutional* architectures as the NN4G of Micheli & Sestito (2005); Micheli (2009).

The neighborhood aggregation is usually implemented using permutation-invariant functions, which make learning possible on cyclic graphs that have no consistent topological ordering of their nodes. A rather general and classical neighborhood aggregation mechanism for node $v$ at layer/step $\ell + 1$ is the following:

$$\mathbf{h}_v^{\ell+1} = \phi^{\ell+1}\Big(\mathbf{h}_v^{\ell}, \ \Psi(\{\psi^{\ell+1}(\mathbf{h}_u^{\ell}) \mid u \in \mathcal{N}_v\})\Big) \tag{4}$$

where $\mathbf{h}_u^{\ell}$ is the node embedding of $u$ at layer/step $\ell$, $\phi$ and $\psi$ implement learnable functions, and $\Psi$ is a permutation invariant aggregation function. Note that $\mathbf{h}_v^0 = \mathbf{x}_v$. For instance, the Graph Convolutional Network of Kipf & Welling (2017) implements the following aggregation, which is a special case of the above equation:

$$\mathbf{h}_v^{\ell+1} = \sigma(\mathbf{W}^{\ell+1} \sum_{u \in \mathcal{N}(v)} \hat{\mathbf{L}}_{uv} \mathbf{h}_u^{\ell}), \tag{5}$$

with $\hat{\mathbf{L}}$ being the normalized graph Laplacian, $\mathbf{W}$ is a learnable weight matrix and $\sigma$ is a non-linear activation function.

## B.2  SOME OSM DEFINITIONS

In order to keep the paper self-contained and to illustrate the different ways OSM has been defined, we briefly review the most common definitions. The following subsection summarizes the most commonly used OSM metrics. Some of these metrics are used in the main text.

Let $D = \text{diag}(d_1, \ldots, d_n)$ be the degree matrix with $d_u = \sum_v A_{uv}$. The combinatorial graph Laplacian is defined as $\mathbf{L} = \mathbf{D} - \mathbf{A}$, with eigenvalues $0 = \lambda_1 \leq \lambda_2 \leq \cdots \leq \lambda_n$. In addition, the symmetric normalized Laplacian, defined by $\hat{\mathbf{L}} = \mathbf{I} - \mathbf{D}^{-1/2}\mathbf{A}\mathbf{D}^{-1/2}$ (assuming $d_u > 0$ for all $u$), reduces the influence of node degrees through symmetric degree normalization. For undirected graphs with nonnegative weights, the eigenvalues of $\hat{\mathbf{L}}$ lie in the interval $[0, 2]$. Also, the multiplicity of eigenvalue 0 equals the number of connected components.

The Dirichlet Energy (Chung, 1997), used to measure OSM in Cai & Wang (2020), is defined as

$$\hat{\text{DE}}(\mathbf{H}^\ell) = \text{Tr}\left((\mathbf{H}^\ell)^T \hat{\mathbf{L}} \mathbf{H}^\ell\right) = \frac{1}{2} \sum_{(u,v)\in\mathcal{E}} \left\| \frac{\mathbf{h}_u}{\sqrt{d_u}} - \frac{\mathbf{h}_v}{\sqrt{d_v}} \right\|_2^2 \tag{6}$$

Note that DE can also be computed using $\mathbf{L}$

$$\text{DE}(\mathbf{H}^\ell) = \text{Tr}\left((\mathbf{H}^\ell)^T \mathbf{L} \mathbf{H}^\ell\right) = \frac{1}{2} \sum_{(u,v)\in\mathcal{E}} \|\mathbf{h}_u - \mathbf{h}_v\|_2^2 \tag{7}$$

Other OSM metrics include those with respect to the norm of node embeddings. For instance, Rayleigh Quotient (RQ) (Chung, 1997), which can be interpreted as DE normalized by embedding magnitude, is defined by

$$\text{RQ} = \frac{\text{Tr}\left((\mathbf{H}^\ell)^T \mathbf{L} \mathbf{H}^\ell\right)}{\|\mathbf{H}^\ell\|_F^2}. \tag{8}$$

Similarly, we can replace $\hat{\mathbf{L}}$ with the normalized Laplacian $\mathbf{L}$ to obtain the Rayleigh Quotient with respect to the normalized graph operator, denoted $\hat{\text{RQ}}$. It was introduced for OSM analysis in Cai & Wang (2020) and adopted by various subsequent works (Yang et al., 2020; Di Giovanni et al., 2023b; Roth & Liebig, 2024; Maskey et al., 2023). RQ captures whether embeddings become *relatively* smoother, independent of magnitude.

Mean Absolute Deviation (Chen et al., 2020a) averages cosine dissimilarity between a node and its neighbors.

$$\text{MAD}_G(\mathbf{H}^\ell) = \frac{1}{n} \sum_{v\in\mathcal{V}} \sum_{u\in\mathcal{N}_v} 1 - \frac{\mathbf{h}_v^{\ell T} \mathbf{h}_u^\ell}{\|\mathbf{h}_v^\ell\| \|\mathbf{h}_u^\ell\|} \tag{9}$$

The smoothness metric SMV (Liu et al., 2020) captures a global node-distance average over all node pairs:

$$\text{SMV} = \frac{1}{n} \sum_{u\in\mathcal{V}} \frac{1}{n-1} \sum_{v\neq u\in\mathcal{V}} \frac{1}{2} \left\| \frac{\mathbf{h}_u}{\|\mathbf{h}_u\|} - \frac{\mathbf{h}_v}{\|\mathbf{h}_v\|} \right\| \tag{10}$$

In the main text, we primarily consider DE and RQ. However, all notions convey the same intuition, loss of discriminative variation, yet, as we show, can disagree in practice.

**Convergence-rate results.** Known theoretical bounds show $\text{DE}(H^k)$ decays exponentially with depth $k$, where the decay rate depends on weight spectra and graph eigenvalues (Oono & Suzuki, 2020; Huang et al., 2020). Such convergence rate results primarily make assumptions on the architecture, weight matrix, and activation functions, and may be altered by skip connections, normalization layers, or simple rescaling such as $2W$ (Roth & Liebig, 2024).

For instance, Cai & Wang (2020) propose a bound on the DE of two consecutive message passing layers (similar bounds found in Oono & Suzuki (2020) and Zhou et al. (2021))

$$\hat{\text{DE}}(\mathbf{H}^\ell) \leq (1-\lambda_2)^2 s_{\max}^\ell \hat{\text{DE}}(\mathbf{H}^{\ell-1})$$

being $s_{\max}^\ell$ the square of the maximum singular value of $W^\ell$, and $\lambda_2$ the second smallest eigenvalue of the Laplacian, i.e., the spectral gap. The proof holds when $s_{\max}^\ell < 1/(1-\lambda_2)$.

In addition, Di Giovanni et al. (2023b) further relate Laplacian eigenvalues and weight spectra to explain whether RQ converges to 0 (collapse) or to $\lambda_{\max}$ (no collapse).

### B.3 SOME OSQ DEFINITIONS

For completeness, we summarize some of the most commonly used metrics in the OSQ literature and their relationships. These quantities mainly measure three different aspects of the graph: (i) *sensitivity/Jacobian* measures that capture how information from a distant node $u$ affects a target node $v$ after $K$ message-passing layers; (ii) *topological bottleneck* proxies such as Cheeger-type cut ratios, graph spectrum or curvature scores; and (iii) *distance-based* quantities such as effective resistance that upper-bound information flow.

### B.3.1 SENSITIVITY

For a $K$-layer GNN let $h_u^{(k)}$ denote the embedding of node $u$ at layer $k$. A first proxy for OSQ is the *Influence Score* of Xu et al. (2018),

$$I(u, v) = \left\| \frac{\partial \mathbf{h}_u^K}{\partial \mathbf{h}_v^0} \right\|. \tag{11}$$

Black et al. (2023) sum these sensitivities over *all* unordered pairs,

$$\sum_{u \neq v \in V} \left\| \frac{\partial \mathbf{h}_u^K}{\partial \mathbf{h}_v^0} \right\|, \tag{12}$$

to obtain a graph-level indicator of how much information is lost.

Di Giovanni et al. (2023a) propose a *symmetric Jacobian obstruction* that removes self-influence and degree bias. They define the Jacobian obstruction of node $v$ with respect to node $u$ at layer $m$ as

$$\tilde{\mathbf{J}}_k^{(m)}(v, u) := \left( \frac{1}{d_v} \frac{\partial \mathbf{h}_v^{(m)}}{\partial \mathbf{h}_v^{(k)}} - \frac{1}{\sqrt{d_v d_u}} \frac{\partial \mathbf{h}_v^{(m)}}{\partial \mathbf{h}_u^{(k)}} \right) + \left( \frac{1}{d_u} \frac{\partial \mathbf{h}_u^{(m)}}{\partial \mathbf{h}_u^{(k)}} - \frac{1}{\sqrt{d_v d_u}} \frac{\partial \mathbf{h}_u^{(m)}}{\partial \mathbf{h}_v^{(k)}} \right), \tag{13}$$

being the extension to the Jacobian obstruction of node $v$ with respect to node $u$ after $m$ layers defined as

$$\tilde{\mathsf{O}}^m(u, v) = \sum_{k=0}^m \left\| \tilde{\mathbf{J}}_k^{(m)}(v, u) \right\|. \tag{14}$$

### B.3.2 TOPOLOGICAL BOTTLENECKS

Many OSQ papers measure topological (structural) bottlenecks using spectral or curvature quantities. Note that here we give an intuition based on the spectral metrics (Arnaiz-Rodriguez et al., 2022; 2025; Karhadkar et al., 2023; Banerjee et al., 2022), but a significant part of the literature uses metrics based on curvature (Topping et al., 2022; Liu et al., 2023a; Giraldo et al., 2023; Nguyen et al., 2023).

First, the topological bottleneck can be measured by Cheeger Constant (Chung, 1997), which is the size of the min-cut of the graph.

$$h_G = \min_{S \subset V} \frac{|\{e = (u, v) : u \in S, v \in \bar{S}\}|}{\min\{\text{vol}(S), \ \text{vol}(\bar{S})\}}$$

A small $h_G$ means one can separate $G$ into two large-volume parts by removing only a few edges, i.e., a severe *topological bottleneck*. Cheeger's inequality links $h_G$ to the spectrum of G:

$$\frac{h_G^2}{2} \leq \lambda_2 \leq 2h_G,$$

where $\lambda_2$ is the second eigenvalue of the normalized Laplacian.

### B.3.3 PAIRWISE DISTANCES

The commute time between two nodes (Lovász, 1993) is defined as the expected number of steps that a random walker needs to go from node $u$ to $v$ and come back. The Effective Resistance between two nodes (Chandra et al., 1989), $R_{uv}$, is the commute time divided by the volume of the graph (Klein & Randić, 1993), which is the sum of the degrees of all nodes in the graph. The effective resistance between two nodes is computed as

$$R_{u,v} = L_{ii}^+ + L_{jj}^+ - 2L_{ij}^+$$

being $\mathbf{L}^+ = \sum_{i>0} \frac{1}{\lambda_i} \phi_i \phi_i^T$ the pseudoinverse of $\mathbf{L}$

Then, some measures derived from this metric can be connected with the topological bottleneck (Chung, 1997; Chandra et al., 1989; Qiu & Hancock, 2007). For instance, the maximum effective resistance of a graph is connected with the Cheeger constant as per $R_{\max} = \max_{u,v \in \mathcal{V}} R_{uv}$

$$R_{\max} \leq \frac{1}{h_G^2}$$

and thus also bounded by the spectral gap as

$$\frac{1}{n\lambda_2} \le R_{\max} \le \frac{2}{\lambda_2}.$$

In addition, the total effective resistance $R_{\text{tot}} = 1/2 \sum_{u,v \in \mathcal{V}} R_{uv}$ is bounded to the spectral gap (Ellens et al., 2011):

$$\frac{n}{\lambda_2} \le R_{\text{tot}} \le \frac{n(n-1)}{\lambda_2}$$

Note that the total effective resistance also equals the sum of the spectrum of $\mathbf{L}^+$ $R_{\text{tot}} = n \sum_2^n 1/\lambda_n$.

### B.3.4    CONNECTING SENSITIVITY AND TOPOLOGICAL DISTANCES

The larger Total Effective Resistance ($R_{tot} = \sum_{(u,v) \in V} R_{uv}$) is, the lower the sum of pairwise sensitivities (Black et al., 2023):

$$\sum_{u,v \in V \times V} \left\| \frac{\partial h_v^{(r)}}{\partial h_u^{(0)}} \right\| \le c(b - R_{tot}) \tag{15}$$

The larger the Effective Resistance is, the higher the Symmetric Jacobian Obstruction (Di Giovanni et al., 2023a):

$$\tilde{\mathsf{O}}^m(u,v) = \sum_{k=0}^m \left\| \tilde{\mathbf{J}}_k^{(m)}(v,u) \right\| \le c\, R_{u,v} \tag{16}$$

### B.4    COMPUTATIONAL BOTTLENECK

Oversquashing can also be seen through the perspective of the message-passing computational graph: each message-passing layer expands the set of nodes whose features can influence a target node. If this *receptive field* grows fast, any fixed-width DGN "squash" many signals into a single vector.

**Receptive Field**    Following Chen et al. (2018) and Alon & Yahav (2021) the receptive field was defined recursively as:

$$\mathcal{N}_v^K := \mathcal{N}_v^{K-1} \cup \{w \mid w \in \mathcal{N}_u \wedge u \in \mathcal{N}_v^{K-1}\} \quad \text{and} \quad \mathcal{N}_v^1 = \mathcal{N}_v \tag{17}$$

which can be also seen as the set of $K$-hop neighbors, i.e., nodes that are reachable from $v$ within $K$ hops The number of nodes in each node's receptive field can grow exponentially with the number of layers $|\mathcal{N}_v^K| = \mathcal{O}\left(\exp(K)\right)$ (Chen et al., 2018). For instance, in a rooted binary tree each layer has exactly $b^{K-1}$ new neighbors, so $|\mathcal{N}_v^K| = 1 + b + b^2 + \cdots + b^{K-1} = \Theta(b^K)$.

**Computational Tree**    When evaluating the actual *computational* graph resulting from message-passing, duplicates matter: a node that appears in several branches of the computation tree contributes multiple times, since each distinct walk contributes a separate message. We therefore use the multiset notation to define the *computational tree* for a node $v$ in Eq. 3, reproduced here for convenience:

$$\mathcal{M}_v^1 := \mathcal{N}_v, \quad \mathcal{M}_v^K := \mathcal{M}_v^{K-1} \uplus \left\{ \biguplus_{u \in \mathcal{M}_v^{K-1}} \mathcal{N}_u \right\}. \tag{18}$$

**Computational Bottleneck**    Therefore, we defined in Def. 4.1 the *computational bottleneck* of node $v$ as the size of the *computational tree*, $|\mathcal{M}_v^K|$, for a given node $v$ and number of message passing layers $K$.

The size of the computational bottleneck (multiset receptive field) at node $v$, can be computed as:

$$|\mathcal{M}_v^K| := \sum_{\ell=1}^K \|A^\ell[v,:]\|_1 = \sum_{\ell=1}^K \sum_{u \in \mathcal{V}} (A^\ell)_{u,v} \tag{19}$$

This definition counts every distinct length-$\ell$ walk from $v$ to any node $u$. Equation equation 19 is exactly the row–sum of the powers of the adjacency matrix; it therefore matches the size of the *computational tree*.

Note that the size of the set-based receptive field corresponds to the support of the multiset $\mathcal{M}_v^K$, denoted $\mathcal{N}_v^K := \text{supp}(\mathcal{M}_v^K)$. Therefore, the multiset size $|\mathcal{M}_v^K|$ is always greater than or equal to the size of the support, $|\mathcal{M}_v^K| \geq |\mathcal{N}_v^K|$, since it accounts for path multiplicity.

In early deep-graph networks literature, Micheli (2009) introduced the idea by using the term "contextual window": deeper layers aggregate exponentially many paths unless skip connections or global pooling curb the growth. The multiset perspective in Eq. equation 19 makes this explosion explicit and the matrix computation directly links to matrix-power interpretations of message passing.

Figure 6 visualises the difference between the set size $|\mathcal{N}_v^K|$ and the multiset size $|\mathcal{M}_v^K|$ on a toy graph.

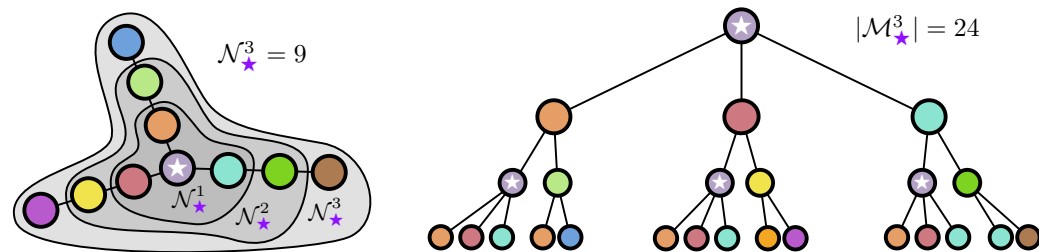

Figure 6: **Computational Bottleneck**. Illustration of a receptive field – defined with sets ($K$-hop neighborhood) – and the definition of computational bottleneck measured as the size of the computational graph – defined with multisets.

In conclusion, we note that in message-passing, the computational bottleneck is driven not by how many distinct vertices are in the $K$-hop neighborhood, but by the size of the computational graph.

## C  SENSITIVITY DECREASES ON A GRID GRAPH WITHOUT TOPOLOGICAL BOTTLENECKS

To show that low sensitivity does not necessarily imply a topological bottleneck, Figure 7 analyzes sensitivity's decreasing trend when the number of message passing layers $L$ increases on the grid graph of Figure 4 of size $10 \times 10$. Increasing the size of the embedding space postpones the collapse of the sensibility.

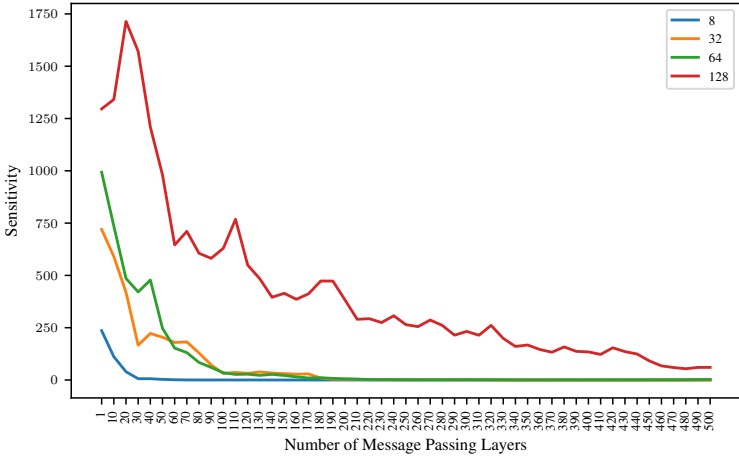

Figure 7: We plot the sensitivity of the grid graph of Figure 4 for the Graph Convolutional Network (Kipf & Welling, 2017) model for different node embedding sizes.

# D    OVERSMOOTHING DOES NOT ALWAYS HAPPEN

For enhanced clarity and to allow for a more detailed examination of the OSM behavior discussed in Section 2, a larger version of Figure 1 is provided in Figure 8.

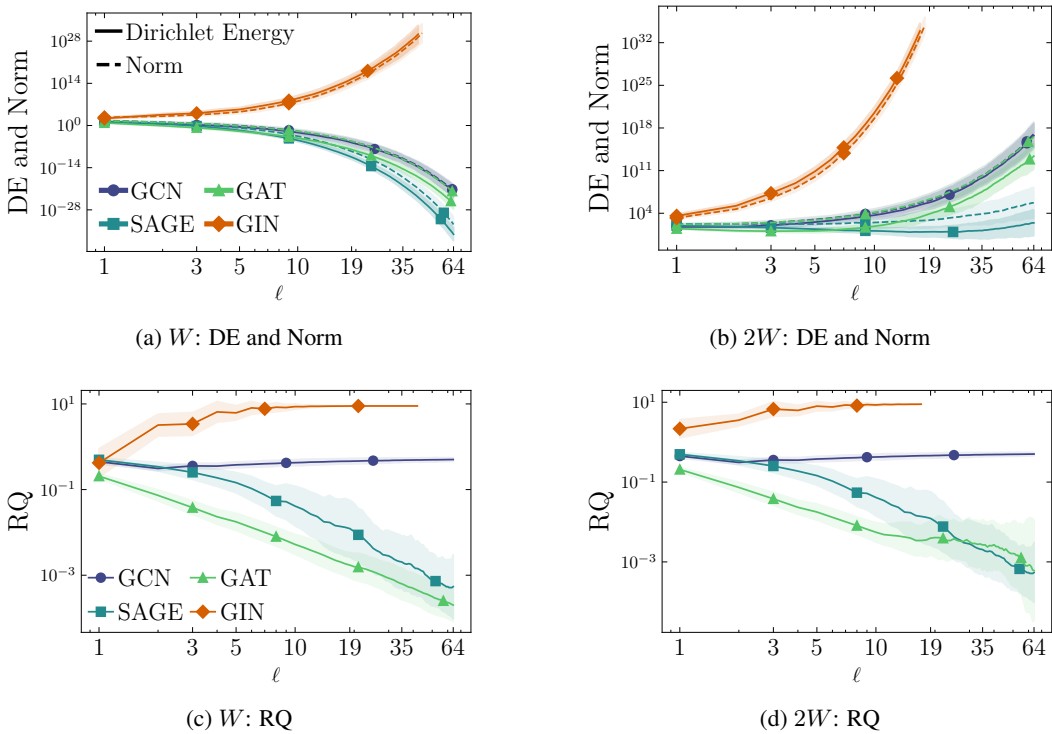

(a) $W$: DE and Norm

(b) $2W$: DE and Norm

(c) $W$: RQ

(d) $2W$: RQ

Figure 8: **Larger version of Figure 1**. **(a-b)**: We depict the evolution, with increasing number of layers, of the DE and the feature norm $\|X\|_F$, using $W$ and $2W$ feature transformations for different architectures. **(c-d)**: Evolution of the RQ for $W$ and $2W$ as before. Experiments run on the Cora dataset for 50 random seeds.

# E    ALTERNATIVE VIEWS

One may argue that the common beliefs of Table 1 could be regarded as alternative views, which we have abundantly discussed and challenged in previous sections. However, we additionally discuss other alternative viewpoints.

One is put forward by Blayney et al. (2025), which states that the oversquashing term is still useful to *describe issues arising from topological bottlenecks and the depth in the computational tree*. The authors argue that all performance issues related to "oversquashing" are caused by either model capacity or low sensitivity, and we do not intend to contest their observations in this space. Their alternative position on the use of the "oversquashing" term likely stems from an in-depth understanding of the distinction between topological and computational bottlenecks as well as the underlying assumptions, which, as this paper shows, cannot be applied to the whole community yet. As a matter of fact, our proposal is pedagogical rather than practical: Section 4 does *not* propose how to identify specific problems (e.g, capacity or sensitivity), but *how to better think* about and approach them without confusing everything under a single umbrella term. For this reason, we believe that a clear separation of concepts, rather than the use of the term "oversquashing", can have a profound impact on the *whole* community in formulating more precise research questions.

Another alternative view is the recent one of Mishayev et al. (2025), which proposes, similarly to Blayney et al. (2025) a problem-oriented dichotomy for the "oversquashing" phenomenon. They argue for the decoupling of this term into computational bottlenecks, which can happen in short-range tasks, and vanishing gradients, which seem to be more related to long-range dependencies. However, because computational bottlenecks can appear in long-range tasks as well as vanishing gradients, we believe this work focuses more on identifying practical problems to address rather than providing a conceptual separation of concepts related to oversquashing. One evidence for this is that the synthetic tasks used in the paper, namely two-radius and ring transfer, contain a topological bottleneck that may have been important to reaching some conclusions. Hence, our goal is complementary to that of Mishayev et al. (2025).

Finally, we mention the position of Kormann et al. (2026). The authors highlight that improvements in over-smoothing metrics and model performance are, at best, weakly correlated. The hypothesis they investigate is that most real tasks rely on short-range dependencies. In addition, they question the focus on curvature in OSQ contributions and call for a more principled understanding supported by appropriate statistics. The authors' views and empirical observations are therefore complementary to ours.

