# OpenReview forum: "Oversmoothing, "Oversquashing'', Heterophily, Long-Range, and more: Demystifying Common Beliefs in Graph Machine Learning"
_ICLR.cc/2026/Conference — ICLR 2026 Poster_

### Official Review · Reviewer_SstW · 2025-10-26

**Soundness:** 3
**Presentation:** 3
**Contribution:** 3
**Rating:** 6
**Confidence:** 4

**Summary:**

This paper clarifies recent concepts in the graph machine learning community, such as oversmoothing and heterophily, and raises simple but noteworthy counterexamples.

**Strengths:**

1. The paper is well motivated as a position paper, covering a wide range of recent trends in graph machine learning.
2. The paper is well structured, clarifying the nine beliefs step by step.

**Weaknesses:**

1. Section 3.1: The example is too specific—showing a heterophilic graph where a DGN can achieve perfect classification. Most examples in this paper are toy problems that can be solved by simple graph functions (degree, distance). These problems don't require DGNs.
2. Some explanations are insufficient. For example, in Section 3.3, the authors argue that distinguishing the task is what really matters. However, there's no further analysis of which graph tasks should consider heterophily/homophily or distance. The argument relies on only two specific examples (degree, distance). As a position paper that may direct the future of this area, a deeper dive into this question is necessary.
3. Line 358: Is it correct to refer to Figure 2? Also, please explain **message filtering** in Errica et al. (2025) and Figure 4 (middle) more clearly, as readers may not be familiar with it.

**Questions:**

1. Could you provide real-world examples of interactions between features, structure, and class labels?

---

> ### Author Response · Authors · 2025-11-21
> **Response to Reviewer SstW**
>
> We would like to thank the reviewer for taking the time to provide the review, for highlighting the well-motivatedness of the paper and the beliefs. We also thank them for spotting the broken reference, which we will fix. We detail our answers to the reviewer’s concerns:
>
> **W1 [toy examples, simple graph functions 3.1].** Counterexamples are the right tool to refute universal claims (they prove “not always”). Fig. 2 (right) is solvable by distance if one knows the ground-truth labeling function. In practice, this function is unknown, which is why learnable operators are used, but the example still refutes the universal belief it targets. We will make this pedagogical intent explicit. Besides, the example was taken from Ma et al. (2022), who support our arguments theoretically and empirically, as it is already explicitly cited in the Figure 2’s caption.
>
> **W2 [insufficient explanations 3.3].** The claim about distinguishing the task is a natural consequence of having refuted specific beliefs, since we argued that labels, structure, and features cannot be seen in isolation. In our opinion, what the reviewer asks us to provide has so far eluded the entire graph ML community, including us, and we respectfully believe it is unfair to consider this a weakness. Rather, what our work provides are novel viewpoints and considerations that will hopefully help researchers study these aspects more in detail, rather than remaining anchored to old convictions.
>
> In the example of section 3.3, the degree and distance tasks are simple but valid counterexamples of why the universal claims of “homophily is good, heterophily is bad” are not always true. In addition to this simple, illustrative, and valid counterexample, we provide previous work that has analyzed this issue in depth and has also partially questioned the validity of this claim: l. 249 (Ma et al. (2022); Errica (2023); Luan et al. (2023); Platonov et al. (2023b); Wang et al. (2024)) and l.280. (Castellana & Errica, 2023; Zheng et al., 2024).
>
> **W3.** Thank you for spotting the typo, it should refer to Fig 4. We will also better explain the message filtering in more detail for the readers less familiar with the recent work.
>
> **Q1 [interactions between features, structure, and class labels].**
>
> Real world applications of this interplay include works on Deep Graph Generation [Zhu et al 2022], Graph Generation [Guo et al 2022], Graph Structure Learning [Zhu et al 2021; Luan et al 2024], or Causality on GNNs [Jiang et al 2023].
>
> Guo, Xiaojie, and Liang Zhao. "A systematic survey on deep generative models for graph generation." IEEE TPAMI 45.5 (2022): 5370-5390.
>
> Zhu, Yanqiao, et al. "A survey on deep graph generation: Methods and applications." Learning on Graphs Conference. PMLR, 2022.
>
>
> Zhu, Y., et al. (2021). A survey on graph structure learning: Progress and opportunities. arXivpreprint arXiv:2103.03036.
>
> Jiang, W., Liu, H., & Xiong, H. (2023). Survey on Trustworthy Graph Neural Networks: From A Causal Perspective. arXivpreprint arXiv:2312.12477.
>
> ----
>
> **Final remarks:** We thank the reviewer again. If any conclusion is still unconvincing, please identify the belief, the specific flaw (section/figure), and a minimal experiment to directly test the disputed assumption. We hope our clarifications sufficiently addressed the doubts of the reviewer; we would appreciate it if the reviewer could consider increasing the score.

---

### Official Review · Reviewer_xBYT · 2025-10-27

**Soundness:** 2
**Presentation:** 3
**Contribution:** 2
**Rating:** 4
**Confidence:** 4

**Summary:**

The submission under review argues that several “folklore beliefs” in graph ML — oversmoothing, oversquashing, homophily vs heterophily, long-range reasoning — are overstated or misinterpreted. It provides counterexamples, simple constructions, and some empirical plots (e.g., 1–64 layer GNNs on Cora with different scalings).

12154_Oversmoothing_Oversquash

 The paper also proposes to retire the overloaded word “oversquashing” and instead talk about “computational bottlenecks” vs “topological bottlenecks,” which it claims are distinct.

**Strengths:**

1. The paper openly challenges sloppy community narratives about depth, heterophily, and “oversquashing,” and reminds readers that degradation in deep GNNs is not the same thing as classic oversmoothing, and that heterophily alone does not doom GNNs.
2. The paper proposes to split “oversquashing” into two ideas: “computational bottlenecks” (information from exponentially many nodes being crammed into a fixed hidden vector); “topological bottlenecks” (structural choke points / narrow cuts in the graph).

**Weaknesses:**

1. The primary claims of this paper overlap significantly with established findings in the GNN literature.
* On Over-smoothing: The observation that removing feature transformations and non-linearities can boost performance in moderately deep GNNs has been demonstrated by prior work, such as SGC [1] and a KDD'22 paper [2]. These studies have already established that performance degradation in models of the depth used in this paper (e.g., 64 layers) is often attributable to feature transformation and non-linearity, rather than over-smoothing alone, which typically manifests in much deeper architectures.
* On Heterophily: The challenges of heterophily have been extensively analyzed. A well-known paper [3] provides a comprehensive treatment of this topic. Unfortunately, the current work does not seem to offer new perspectives or insights beyond what was presented in [3].
------
2. Given that the paper addresses topics with a rich history of research, a deeper investigation from novel perspectives or a rigorous theoretical analysis would be expected. However, the current analysis lacks the necessary depth.
* The investigation into over-smoothing is confined to a few simple settings on a single dataset (Cora). This limited experimental setup is not convincing enough to reveal generalizable patterns or provide robust evidence for the claims made.
* The discussion on heterophily relies on reusing a toy example from [3] without introducing new conceptual or empirical contributions. A more compelling argument would require novel experiments or theoretical formulations.

------
3. About the statement ‘if different classes imply different feature distributions, why would one need a DGN rather than a simple MLP?’

I also disagree with the way that’s phrased in the submission.
Even if different classes have different feature distributions, classification boundaries may still be ambiguous (e.g., overlapping Gaussians, noisy features, etc.). A plain MLP on node features alone may misclassify nodes in that overlapping region.
A message-passing layer can still add value by smoothing / denoising via neighbors: even if node i’s own features are ambiguous, aggregating its neighbors’ features can push it toward the correct side of the decision boundary. This is exactly the classic semi-supervised “feature smoothing” or “label propagation” intuition behind GCNs.

So “why would you ever need a GNN if features are already somewhat class-dependent?” is a false dichotomy. You still may want graph convolution to regularize local decision boundaries and reduce local noise — especially when each node alone is borderline but its neighborhood is consistent.

----
4. Difference between “computational bottleneck” and “topological bottleneck” is unclear. As a core concept to support your claims, it is not wise to put it to appendix. The main text should be self-contained.

**Questions:**

See weaknesses.

---

> ### Author Response · Authors · 2025-11-21
> **Response to Reviewer xBYT: Part 1**
>
> We thank the Reviewer for recognizing the importance of proposing two separate definitions for the oversquashing problem and for critically evaluating established beliefs in the graph ML field. Below, we address the concerns raised and refer the Reviewer to the general comment for further discussion of their validity. Additionally, we would like to ask the Reviewer to provide references for “[1][2][3]”, which are not included in the review. Without them, we cannot properly respond to each point or add any missing references to the paper.
>
> **W1.**  We would like to clarify that we do not think we made such a claim. If the reviewer believes we do, please point us to the exact line. Our point is that changing architecture/normalization/metrics can decouple OSM from accuracy, contrary to popular belief; we explicitly state OSM is not the sole cause (and not equivalent to accuracy loss) and references [1] and [2] actually support our statements about OSM. We have included them in the paper, thank you for raising them to our attention. However, let us clarify that the goal of the paper, as explicitly stated, is not to provide new insights beyond those in specific works, but rather to leverage this knowledge to demystify common beliefs. We are aware that some expert authors have an in-depth knowledge of specific problems, for instance heterophily/homophily, and are able to distinguish popular misconceptions, but the scope of our work is to make explicit, in a single paper, all beliefs that are affecting our field and support our arguments with simple but sufficient counterexamples. We believe there is value in this kind of work.
> - In particular, we are aware of [3], however we respectfully argue we did not simply took inspiration from [3] for Figure 2 (left); indeed, we brought to light that the considerations in [3] rely on an **implicit assumption** that is not made explicit in the text, and that assumption corresponds to the belief of Section 3.2. The results of [3], including the very useful correlation between cross-class neighborhood similarity (CCNS) matrix and performance, may not hold anymore under a different assumption It is important to highlight how such assumptions can deeply affect our conclusions, and to the best of our knowledge we are the first to clearly highlight this and put it in relation with other common graph ML beliefs.
>
> **W2.** Counterexamples are designed to be minimal in order to challenge universal claims; when more extensive theories or empirical evidence exist, we refer to them. Our main contribution is identifying the beliefs in the literature, synthesizing and defining boundaries: documenting how common they are, clarifying what follows and what does not (by refuting by counterexample), and laying down the distinction between computational bottlenecks and topological ones. If the reviewer believes our reasoning has flaws, please specify which belief remains unrefuted or unaffected, the specific flaw (section/line), and a targeted experiment that could settle it. We are happy to provide such an experiment. However, we hope to have clarified the scope our our paper more extensively at this point.
>
> We direct the reviewer to the answers to W2 and W3 in reviewer 348B for additional clarification on the scope and adequacy of our analysis, as well as to the answer to W1 from reviewer R4Aqk regarding the specific example refuting the OSM claims.
>
>
> **W3.** We completely agree with the reviewer, which is exactly why lines 274-276 recite “Note
> that this discussion is irrespective of the denoising/regularizing effects of DGNs in semi-supervised scenarios compared to MLPs Hoang et al. (2021); Errica (2023).” Our argument was meant to be slightly provocative, but we agree that we should phrase it better to explain exactly that the interplay between labels, feature, and structure can still matter even in the context being discussed.
>
>
> **W4.** Thank you to the reviewer for pointing out this question, which will help us highlight an aspect of our work that was previously overlooked. We actually do have formal definitions of both concepts in the paper.
> - First, we define the computational bottleneck in Appendix B4, where we formalize it in Definition B.1, built upon the “contextual window” introduced by Micheli (2009). We also provide the main difference between the classical computation of the receptive field, commonly used in GNNs, and why the computational bottleneck should be measured with supersets. Although the exact derivations are provided in the appendix, this is referenced in the main text (l. 334).
> - Second, we already defined in Appendix B.3.2 that the topological bottleneck is measured by the spectral gap of the curvature. In the literature, there is no unique way to formally define a topological bottleneck, so it is rather defined in terms of proxy metrics.

---

> > ### Comment · Reviewer_xBYT · 2025-11-21
> > **Providing references**
> >
> > Hi,
> >
> > Thanks for the rebuttal. I haven’t had the chance to read it thoroughly yet. I also apologize for forgetting to provide the references earlier—please find them below:
> >
> > [1] Wu et al., “Simplifying Graph Convolutional Networks,” ICML 2019
> >
> > [2] Zhang et al., “Model Degradation Hinders Deep Graph Neural Networks,” KDD 2022
> >
> > [3] Ma et al., “Is Homophily a Necessity for Graph Neural Networks?” ICLR 2022

---

> > ### Author Response · Authors · 2025-11-23
> > **Response to Reviewer xBYT: Part 2**
> >
> > Although we provide the derivations, we acknowledge that the reference to the appendices in the main text could be improved. We will include these sections when introducing the topological versus computational bottlenecks in the main text, using the extra page allowed, and leave the expanded parts for the appendix details. We will also change Figure 3 to explicitly account for this change. In addition, if any part of the formalism is unclear, we would appreciate it if the reviewer could let us know what is missing so we can clarify or improve the explanation.
> >
> > ----
> >
> > **Final Remarks:** We thank the reviewer again for the time spent on the review, and we hope to continue the discussion to clarify the paper's scope and technical aspects. Please let us know if our answers address your doubts. If not, we are happy to further delve into specific points. We hope the reviewer will consider increasing the score in light of our clarifications.

---

> > > ### Author Response · Authors · 2025-11-23
> > > **Response to Reviewer xBYT: Part 3**
> > >
> > > We appreciate the reviewer’s prompt answer. **We have revised our previous rebuttal based on the provided references.** In summary:
> > > ' We did not think we made the claim stated by the reviewer, but references [1] and 2 are further evidence in support of our statement. Note that the goal of our work is to raise awareness about a number of beliefs, and we support our statements with previous empirical and theoretical knowledge.
> > > ' Compared to [3], we brought to light that the results rely on an implicit assumption, which can lead readers to wrong conclusions if the assumption is not made explicit and carefully taken into account. We are the first to highlight this issue.
> > >
> > >
> > > We did our best to comment on the provided reference as soon as possible, so the reviewer will have the full picture when they will read our rebuttal.

---

### Official Review · Reviewer_4Aqk · 2025-11-03

**Soundness:** 3
**Presentation:** 3
**Contribution:** 2
**Rating:** 4
**Confidence:** 4

**Summary:**

The paper analyzes nine common beliefs in Deep Graph Networks concerning issues such as over-smoothing, over-squashing, homophily/heterophily, and long-range tasks. It systematically elucidates their origins, identifies misconceptions through experimental validation and counterexamples, and aims to correct erroneous understandings, thereby contributing to the advancement of the graph learning field.

**Strengths:**

S1. The article systematically examines prevalent misconceptions and imprecise definitions within the field of DGN. It offers a reflective analysis aimed at clarifying commonly misunderstood concepts such as over-smoothing and over-squeezing, thereby facilitating researchers' comprehension and preventing misinterpretations. This critical review encourages ongoing reflection on entrenched perspectives, fostering a more rigorous and accurate understanding in the domain.

S2. The paper refines the understanding of the "over-squashing" problem, elucidating the relationships among over-smoothing, over-squashing, homophily/heterophily, and long-range tasks. By redefining these issues and their key aspects, the discussion aims to promote advancements in the development of DGN methodologies.

**Weaknesses:**

W1. In refuting the claim that OSM is a property of all DGNs, the article relies solely on a simple experiment, which lacks rigorous proof and suffers from limited experimental evidence. The experiment, on one hand, uses only the Cora dataset, thereby failing to establish the universality of the conclusion; on the other hand, certain inferences merely suggest that the DE and RQ metrics, which reflect smoothing, are insufficiently convincing, yet they do not demonstrate that OSM is inevitably unavoidable. It is recommended that the authors provide more rigorous theoretical justification or more comprehensive experimental evidence when discussing the underlying causes of OSM.

W2. The definitions of "computational bottleneck" and "topological bottleneck" rely primarily on intuitive explanations, lacking a unified mathematical framework or quantifiable metrics.

**Questions:**

Q1. The paper suggests that research should focus on node separability rather than OSM metrics. Could you provide a novel, quantifiable "node separability metric" as an alternative?

Q2. Refining the tasks of distinguishing "computational bottlenecks" from "topological bottlenecks" is meaningful. I hope the authors can further discuss whether there are actual examples of such distinctions in real-world graph data, whether these distinctions are feasible, and how to differentiate them: is it possible to propose theoretically distinguishable metrics?

Q3. Do the conclusions drawn in the paper remain valid in the context of Transformer-based GNNs or Graph Diffusion Models?

Q4. The paper highlights that certain common misconceptions lack universality. Could the authors further elaborate on the scope of applicability for traditional beliefs? For example, in Section 3.1, the counterexample "if a node is at a distance greater than five from a specific node" relates solely to the topological properties of the graph, independent of homophily or heterophily. Under what circumstances is it meaningful to discuss homophily and heterophily?

---

> ### Author Response · Authors · 2025-11-21
> **Response to Reviewer 4Aqk: Part 1**
>
> We appreciate the Reviewer’s recognition of the importance of proposing two distinct definitions for the oversquashing problem and critically examining established beliefs in the graph ML field. We also thank you for raising interesting questions that may lead to future research ideas. Below, we address the concerns raised and refer the Reviewer to the general comment for further discussion of their validity.
>
>
> **W1 [simple experiment in OSM always happens].** Universal claims, such as the one listed in Table 1,  are typically refuted by **counterexample**; that is the correct standard. We opted for a simple but realistic experiment because it is not necessary to do something more complex, and something more complex would not serve the purpose of being easy to remember. The experiment demonstrates that not all architectures exhibit OSM by default, and that small tweaks to the training and evaluation procedures (such as changes to architecture, weights, or metrics) can yield different results in the OSM analysis. The example is sufficient to show that OSM does not always occur. If it does not happen in this case, it is enough to say that it does not always happen.
>
> Summarizing, we are refuting that general claim by finding a counterexample, and **finding a counterexample is enough to disprove the claim**. If the example is simple, it would not only improve the exposition but also enhance understanding.
>
> In addition, the paper already builds on a more rigorous theoretical justification and comprehensive experimental evidence, properly citing references that analyze different individual perspectives on this problem. (Zhang et al., 2025) (Keriven, 2022; Roth & Liebig, 2024; Wu et al., 2023b)  (Zhao & Akoglu, 2020; Cong et al., 2021; Yang et al., 2020; Arroyo et al., 2025; Park et al., 2025).
>
> If the reviewer believes a counterexample is invalid or inadequate, we kindly ask to point us to its specific limitations so we can further understand your concerns and revise the paper accordingly. We hope, however, to have provided more context for why we only need simple but effective counterexamples.
>
> **W2 [no formal definition of 2 facets of OSQ].** Thank you to the reviewer for pointing out this question, which will help us highlight an aspect of our work that was previously overlooked. We actually do have formal definitions of both concepts in the paper.
> - First, we define the computational bottleneck in Appendix B4, where we formalize it in Definition B.1, built upon the “contextual window” introduced by Micheli (2009). We also provide the main difference between the classical computation of the receptive field, commonly used in GNNs, and why the computational bottleneck should be measured with supersets. Although the exact derivations are provided in the appendix, this is referenced in the main text (l. 334).
> - Second, we already defined in Appendix B.3.2 that the topological bottleneck is measured by the spectral gap of the curvature. In the literature, there is no unique way to formally define a topological bottleneck, so it is rather defined in terms of proxy metrics.
>
>
> Although we provide the derivations, we acknowledge that the reference to the appendices in the main text could be improved. We will include these sections when introducing the topological versus computational bottlenecks in the main text, using the extra page allowed, and leave the expanded parts for the appendix details. We will also change Figure 3 to explicitly account for this change. In addition, if any part of the formalism is unclear, we would appreciate it if the reviewer could let us know what is missing so we can clarify or improve the explanation.
>
>
> **Q1 [node separability metric].** This is an interesting question that we actually suggest exploring as a future direction in line 194. More specific ideas could combine the Group Distance Ratio (Zhou, 2020), which measures the inter-group distance relative to the intra-group distance in Euclidean space, with common spectral (DE, RQ) or rank collapse metrics. Additionally, for label-aware OSQ, potential future metrics could build upon what Wang and Leskovec (2021) proposed, a type of Jacobian label-aware sensitivity, which was not originally proposed for OSQ. Another possibility is to use a divergence measure between node distributions modeled as in Errica (2023). All these citations are already included in the paper.

---

> > ### Author Response · Authors · 2025-11-21
> > **Response to Reviewer 4Aqk: Part 2**
> >
> > **Q2 [tasks-examples-math 2 facets].** Thank you for the suggestion: we will clarify that topological bottlenecks are intrinsic to the graph (related to narrow cuts/curvature, spectral criteria, and the Cheeger constant), while the computational bottleneck is intrinsic to the architecture or procedure (such as expanding the computational tree to a fixed capacity). While there are already metrics for topological bottleneck, the size of the computational tree could be another way to measure it. In the paper, we outline examples (Figs. 3, 4, and 5) and explain in which cases both bottlenecks appear or just one of them. In the real world, graphs with hubs (Fig. 5) may correspond to social networks, while examples like grids could correspond to power-grid networks or even attention matrices in LLMs, where oversquashing (the computational bottleneck facet) has also been identified (Barbero, 2024).
> >
> > We also remind the reviewer that the mathematical and formal description is already introduced in the paper: computational bottlenecks (formalized in App. B.4, Def. B.1) and topological bottlenecks (measured via the spectral gap of curvature in App. B.3.2). We’ll incorporate brief definitions directly into the main text to enhance its self-contained nature.
> >
> > Barbero, F. et al. NeurIP S2024.  Transformers need glasses! Information over-squashing in language tasks
> >
> >
> > **Q3 [TGNN & DGM].** We appreciate the reviewer's interesting question. This inquiry is beyond the scope of our paper, which focuses solely on the problems of DGNs, particularly MPNNs. Addressing this question would require a completely separate analysis.
> >
> > However, out of curiosity, the concept of over-squashing in transformers has already been studied in Barbero 2024, in graph transformers in Muller (2023), and in spatiotemporal GNNs by Marisca (2025).
> >
> > Müller, L., Galkin, M., Morris, C., & Rampášek, L. (2024). Attending to graph transformers. TMLR
> > Marisca, I., Bamberger, J., Alippi, C., & Bronstein, M. M. (2025). Over-squashing in Spatiotemporal Graph Neural Networks. NeurIPS 2025.
> >
> > **Q4.** As the reviewer suggests, the task in S3.1 shows that it is independent of homophily/heterophily. That counterexample suggests that even a mostly homophilic graph can be very difficult to classify, and vice versa. Therefore, we claim that homophily/heterophily is a function of the task, but the converse is not true; thus, it’s more meaningful to focus on the task and embedding separability rather than on just the heterophily and homophily of the graph.
> >
> > ----
> >
> > **Final remarks:** We thank the reviewer again for allowing us to clarify the concerns about the paper and for suggesting interesting ideas for future work. We are happy to continue this fruitful discussion and we hope to have clarified the points above. We would sincerely appreciate it if the reviewer could consider increasing the score for our work given our clarifications.

---

### Official Review · Reviewer_348B · 2025-11-03

**Soundness:** 1
**Presentation:** 2
**Contribution:** 3
**Rating:** 2
**Confidence:** 4

**Summary:**

This paper revisits and critiques nine widely discussed beliefs in the field of Graph Machine Learning (GML), covering issues such as over-smoothing (OSM), over-squashing (OSQ), homophily/heterophily, and long-range dependency. Through a series of counterexamples, the authors reveal common misconceptions, for instance, that over-smoothing is not inevitable; it depends on model architecture, hyperparameters, and evaluation metrics, and bears no direct causal relation to performance. The paper aims to clarify these misunderstandings and calls for a more rigorous research in graph learning.

**Strengths:**

1. The paper’s central motivation—to critically examine and challenge long-held opinions and assumptions in the GML community—is very interesting and valuable.

2. The writing is clear and well-structured, using intuitive examples and accessible language to convey complex ideas effectively.

**Weaknesses:**

1. Lack of justification for the representativeness of rebutted claims. The paper does not convincingly establish whether the criticized viewpoints truly represent mainstream consensus in the field. For instance, is the claim that “all GNNs inevitably suffer from over-smoothing” genuinely a widely accepted belief, or is it only mentioned in a limited subset of works? To make the critique compelling, the authors should first demonstrate the prevalence and influence of these viewpoints in prior literature.

2. Insufficient theoretical and empirical validation. The arguments rely heavily on qualitative reasoning and illustrative examples, with very limited mathematical formalism or empirical validation. I am not convinced. To strengthen credibility, the paper should incorporate formal theoretical derivations, quantitative experiments, or counterfactual analyses that can substantiate its claims.

3. The manuscript attempts to address too many issues simultaneously, resulting in a lack of depth and weakened persuasiveness. I strongly recommend focusing on one or two truly dominant misconceptions and providing a rigorous, well-supported, and empirically validated analysis of these specific points. For example, the classic paper “Adversarial examples are not bugs, they are features” offers a good example: it concentrates on a single misconception and dismantles it thoroughly through both theoretical modeling and empirical evidence.

4. Several arguments (e.g., regarding differences between homophilic and heterophilic graphs) have already been discussed—explicitly or implicitly—in prior works. For example, the development of heterogeneous graph neural networks (HGNNs) stems from recognizing that aggregation mechanisms in homophilic graphs are inadequate for heterophilic structures.

**Questions:**

Please see the weakness.

---

> ### Author Response · Authors · 2025-11-21
> **Response to Reviewer 348B: Part 1**
>
> We thank the Reviewer for recognizing the value of critically challenging common beliefs in the graph ML field. Below, we address the weaknesses raised by the Reviewer and refer the Reviewer to the general comment for further discussion on the appropriateness of the concerns.
>
> **W1 [Lack of justification of claims].** We have already established prevalence and influence: for “OSM is a property of all DGNs,” we list 36 works ($\sim$17k citations); for “OSM causes performance degradation,” 30 works ($\sim$16k citations). This is sufficient to show the beliefs are mainstream and impactful. If you believe this is not representative, please list the specific high-impact papers you think we missed and explain how including them would reverse the prevalence conclusion.
>
> **W2 [Insufficient theoretical and empirical validation].**
> Universal claims, such as the one listed in Table 1, are **refuted by counterexample**; that is the correct standard. Where full theory/empirics already exist, we cite them (e.g., lines 361-377); where needed, we provide minimal experiments (e.g., Fig. 1 and Fig. 7), decisive constructions (e.g., Figs. 3, 4, and 5), and references to formal results (e.g., “beneficial smoothing” from Keriven 2022 or “rank collapse” from Roth 2024). For OS,Q we also provide formalism, as we define computational vs topological bottlenecks and give metrics/derivations (App. B.4, B.3.2).
>
> We want to highlight to the reviewer that these nine beliefs did not exist or were not previously identified systematically. The primary contribution of this work is the identification of these beliefs through a systematic literature review, together with the reuse of known theoretical results to support our arguments. Furthermore, we analyze all beliefs in detail, providing simple, realistic examples, and outline future research directions for each of the nine.
>
> We believe our paper already provides credible evidence. The reviewer did not question the validity of our claims but rather the lack of complexity, which, as we argue above, is not strictly necessary to refute universal claims. To make your comment actionable, we would appreciate it if the reviewer could specify which belief remains unrefuted, which assumption in our argument is incorrect (cite section/line), and what experiment (dataset, model class, metric, hypothesized result) would address that flaw.
>
> **W3 [too many issues simultaneously - lack of depth].** Breadth is pursued with a purpose: the beliefs are interconnected, and our role is to demystify common, largely known and accepted beliefs, to document prevalence, to reveal hidden assumptions, to challenge universal approaches, and to guide future research. We believe there has been a misunderstanding around the true purpose of our work: there are already prior works, which we cite, that support our individual arguments with rigorous, well-supported, and empirically validated analyses, and the goal of this work is exactly to put them together, something which no one did before us to provide a clean, unambiguous and unified picture of the different beliefs to the community. **We believe there is value and utility in this**, regardless of the presence of novel theoretical derivations, methods, and quantitative analyses.
>
> If you feel depth is lacking for a particular belief, we would appreciate it if the reviewer could indicate the missing analysis or experiment that would change your verdict alongside the specific reason for it. However, we hope that our clarification around the purpose of our work has shed more light on our specific choice of presentation.

---

> > ### Author Response · Authors · 2025-11-21
> > **Response to Reviewer 348B: Part 2**
> >
> > **W4 [arguments in prior work].** As stated in the introduction, this work presents a critical analysis of the literature: our contribution is to systematize popular beliefs and clarify what follows and what does not across OSM/OSQ/heterophily/long-range, exploiting recent theoretical and empirical evidence that is not yet broadly accepted nor known by the community at large.
> >
> > The significant contribution lies in identifying and making explicit common beliefs (by number of publications and impact), some of which were previously mentioned partially and briefly in previous work (e.g., beneficial smoothing was mentioned in Keriven 2022, which is only a subset of the “OSM is the cause of performance degradation”  belief). We are fair with the credit attribution. For instance, in the belief “OSM is a property of all DGNs.” we mention  Zhang et al 2025 or Cong et al 2025 (l121 and l144); an in the belief “OSM is the cause of performance degradation” we mention works on "beneficial smoothing” (l197) and works on “feature collapse not only due to osm” (l 204).
> >
> > Regarding the specific reviewer's concern about HGNNs designed for heterophily, we are not entirely sure what point they are trying to make. Are they implying that HGNNs were developed under the assumption that GNNs are primarily for homophily? That would precisely be one of the high-level assumptions mentioned in that section, which is composed of more fine-grained assumptions (eg, “homophily is good and heterophily is bad”) that we challenge in Section 3 by showing counterexamples and pointing to some previous literature challenging that assumptions. Many studies have critiqued this assumption from different angles: whether we measure homophily accurately, whether we evaluate it correctly, and whether even common GNNs can handle certain types of heterophily: l. 249 (Ma et al. (2022); Errica (2023); Luan et al. (2023); Platonov et al. (2023b); Wang et al. (2024)) and l.280. (Castellana & Errica, 2023; Zheng et al., 2024). We would appreciate it if the reviewer could clarify the reference to HGNNs.
> >
> > ----
> >
> > **Final remarks:** We sincerely hope our answers better clarify the paper's goal and, most importantly, the adequacy of our analysis given this scope. If the reviewer remains unconvinced, we would like to request more information on why the lack of theoretical derivations and quantitative analyses invalidates our arguments, as they are already based on previous work. We appreciate any clear suggestions for improvement that point to specific missing aspects of a given belief. We would appreciate it if the Reviewer could consider raising the score in light of our answers.

---

### Author Response · Authors · 2025-11-21
**General Comment to Reviewers: Part 1**

We would like to provide a general comment to all reviewers, whom we deeply thank for their efforts. **We have uploaded a revised PDF version of the manuscript.** Below, we provide a general answer for the reviewers summarizing our clarifications.

We understand that this paper do not align with classical mainstream contributions of current research papers (new technical method and experiments), but we believe that the level of technical contribution required to conduct this analysis is high, and that the analysis performed is of high significance for the community (identify clear beliefs in the literature + show clear counterexamples + provide future lines of research). We believe that it is crucial to better clarify the scope of our work, before commenting on the concerns raised by the reviewers. We hope the reviewers will do the same with our answers.

Thank you again and we hope to engage into a constructive and meaningful discussion together.

**_Scope_**

We appreciate the feedback and would like to clarify the paper's goal. Our aim is to raise the standard of conceptual clarity in GML by identifying, across a broad body of literature, nine prevalent beliefs, making their underlying assumptions explicit, and disproving **universal claims through counterexamples**, while pointing to prior rigorous theoretical and empirical results that are available but not as well-known. For beliefs stated as universal claims, a single valid counterexample suffices to refute the universal form. **Counterexamples effectively challenge universal claims, do not need to be complex, and are sufficient for refutation, which is intentional** and fundamental to our contribution.



Specifically, our paper offers a critical analysis: we (i) examine the assumptions behind each belief, (ii) disprove or qualify them using counterexamples and references to previous theoretical or empirical findings, and (iv) propose future research directions. This contribution does not require introducing a general method or benchmark to be of value to the graph machine learning community, which is the ultimate aim of this work. We hope the reviewers will acknowledge this.
We also reckon the paper is timely and helpful, as it prevents misleading narratives (e.g., “OSM leads to accuracy drop”) from further spreading, thus steering the community's thinking and effort towards more impactful and solid directions.

**_Representativeness_**

Regarding the representativeness of the beliefs, for each claim, we compile numerous highly-cited papers that support or rely on it (e.g., 36 works supporting “OSM is a property of all DGNs” with about 17k citations; roughly 30 works on “OSM causes performance drops” with approximately 16k citations). We also cite sources that question these beliefs, demonstrating community engagement and importance. This is the core prerequisite for a demystification paper, and we believe we have met it.

**_“Already known results”_**

Precisely because some of the beliefs are partially mentioned in some prior work, the field-level synthesis and boundary-setting are new and valuable. The reviews do not identify a prior work that (i) enumerates these nine beliefs, (ii) documents their prevalence, and (iii) systematically separates the valid core from the overstated universal claim by providing counterexamples. If the contributions now seem obvious to the reviewers, it is because we have achieved our goal of disseminating these concepts altogether. However, we have reasons to believe, by talking to many expert colleagues, that these concepts are not generally obvious nor well-known at all.


**_Evidence, technical contributions, and insufficient validation_**

We understand that some reviewers have complained about the absence of novel technical or quantitative analyses, but this is a different kind of work.. Once more, the main contribution corrects universalized technical folklore. For such claims, **refuting the universal claims by counterexamples and references to existing theories and experiments is an appropriate strategy**. We have always included rigorous references and simple counterexamples to clarify assumptions, along with quantitative experiments and formalizations where needed. We cannot report all the theoretical conclusions from previous work in this paper, so we cite them to give readers the opportunity to dive deep into a specific topic. Additionally, the paper offers an important theoretical contribution by proposing to distinguish two bottlenecks in Oversquashing: a computational one related to receptive field growth and a topological one related to graph structure. Following R4Aqk's suggestion, we moved these definitions to the main paper and properly cited the appendix.

---

> ### Author Response · Authors · 2025-11-21
> **General Comment to Reviewers: Part 2**
>
> **_Clarifications on heterophily_**
>
> Reviewer 348B claims that the development of GNNs stems from recognizing the limitations of MPNNs on heterophilic graphs, which is the main simple assumption of the entire field of heterophilic GNNs. In that field, several fine-grained assumptions have been made, which are precisely the ones that we criticize throughout Section 3. Again, we use explicit references to literature or counterexamples to refute the general claims.
>
> For instance, our use of simple counterexamples intentionally pins down task-label mechanisms (such as degree and distance) and demonstrates that broad claims (“heterophily dooms GNNs”) rely on implicit task assumptions and specific metrics. We expand Sec 3.3 with guidance: when heterophily affects decision-making (for instance, when the target depends on neighborhood labels versus features or label/noise regimes).
>
> ----
>
> **Final remarks:** Although we provide this general clarification, we believe most of the reviewers' concerns about the tone, scope, and contributions are clearly addressed in the paper. For example, Reviewer 348B's concerns about the heterophilic section are thoroughly covered throughout section 3, and concerns about the representativeness of the citations should be resolved by the large number of citations for each belief and their significance.
>
> Finally, in light of what we said and to address the reviewer’s specific concerns, it would be important to us that the reviewers were more specific about their concerns and indicate exactly where our analysis is flawed, as well as which specific experiment would resolve the issue. For each point, it would be very helpful to provide (1) the belief you think remains unrefuted, (2) the precise flaw in our reasoning (e.g., an unstated assumption, an invalid counterexample, cite section/figure/line), and (3) a concrete suggested counterexample or experiment, and the hypothesized outcome that would fix the flaw of our analysis. We consider statements like “insufficient validation” or “too broad” hard to work with.

---

### Meta-Review · Area_Chair_3L4p · 2026-01-07

**Summary:**

This paper has mixed borderline reviews. The main concerns of the reviewers are that 1) the scope of the paper is broad and thus it is not possible to dive deep into the details for each issue, 2) there already exists  literature refuting some of the beliefs, 3) the counterexamples are simple, 4) the paper does not propose solutions or new algorithms (but it does propose a solution to the issue about the ovesquashing terminology). However, in the paper and in their rebuttal, the authors clearly outline the goal and the scope of the paper and explain why they believe that the goal is achieved.

In my opinion, the paper brings up a very important topic. Indeed, commonly accepted beliefs and assumptions around such popular concepts as oversmoothing, oversquashing, and the homophily-heterophily dichotomy are not always true but significantly affect current research in graph ML. This paper collects all these common beliefs and assumptions and provides simple and intuitive explanations, counterexamples, or experiments. The authors also cite related work discussing or refuting some beliefs where available. I believe that this paper will be useful for the community and will direct further research towards well-thought-out and properly formulated concepts and problems. It can also serve as a reference for graph ML research and as a good overview citing a lot of existing papers that discuss these subjects. While the paper may have certain weaknesses, I believe that its potential positive effect outweighs its limitations.

I advise the authors to consider the following points when preparing a revised version of this paper:
- Please, proofread the text. The current text contains some typos (e.g., repeating words like "neighbors neighbors"). Also, when formatting citations, \citet vs \citep are incorrectly used in multiple places.
- Regarding the discussion about homophily/heterophily, I think it's also worth raising the terminology issue. Namely, different papers define 'homophily' differently. The standard way to (informally) define homophily is "when similar nodes are connected". As discussed in the paper, this can relate to class labels, node features, or both. However, there are papers that propose new measures that better correlate with GNN performance. Some of them refer to the new measures as *alternatives* to homophily while other refer to them as *new better homophily measures*. Thus, in the literature, homophily sometimes is a synonym to "a measure that correlates with GNN performance," which may also cause confusions in the literature on homophily and heterophily.
- If possible, please add more details for the experiment in Figure 1. Namely, what difference does the rescaling make? Do we obtain the same trained models (but with scaled weights) after the training? What is the intuition behind the obtained different behavior? If I understand correctly, the referred paper by Roth & Liebig uses random weights for a similar experiment.
- Minor point: the title of Section 4.3 can be understood not as intended. As I undestand, this section claims that oversquashing *is* problematic for long-range tasks, but not only for them.

**Reviewer Concerns:**

There are a number of concerns raised by the reviewers, some of them a listed below.

There exists previous literature on the topics discussed above; thus, novelty is limited. The authors replied that the goal of this paper was to combine all the (related) beliefs in one work. They also properly cite the related papers.

The coverage is too broad to dive deep into the details for each issue. The paper does not contain deep theoretical or empirical analysis. Here, the authors explained that the goal was to cover many typical beliefs in one work, and they intentionally provided simple and intuitive counterexamples or cited the related work where possible.

Examples are too simple. The authors reply that the examples are intentially as simple as possible since it is suffient to demostrate their points.

The paper does not propose solutions to some of the problems. The authors argue that this is out of the scope of this paper.

**Reviewer Scores:**

The original scores are 4 4 6 2. In my opinion, the authors have reasonably addressed the raised concerns. In particular, they outline the scope and purpose of this work. Unfortunately, the reviewers did not engage in the discussion (while it was still open), and it is hard for me to predict whether the review scores would changed based on the authors' rebuttal.

---

### Decision · Program_Chairs · 2026-01-26

Accept (Poster)